# Riemannian Diffusion Models on General Manifolds via Physics-Informed Neural Networks

**Gyeonghoon Ko** [1]   **Juho Lee** [1]

## Abstract

Riemannian diffusion models generalize score-based generative modeling to manifold-supported data via stochastic diffusion equations on the manifold. However, training requires sampling from and differentiating the manifold heat kernel, which is rarely available in closed form beyond a few highly symmetric manifolds. We propose a general approach that approximates the heat kernel by directly solving the manifold heat equation with a physics-informed neural network (PINN). Given an explicit manifold specification, we choose a coordinate system, derive the corresponding heat (Fokker–Planck) equation and a short-time asymptotic approximation, and then train a PINN to learn the log heat kernel. The resulting surrogate enables both forward noising (heat-kernel sampling) and conditional-score evaluation for denoising score matching. We demonstrate the method on diverse manifolds including $S^2$, $SO(3)$, $\mathrm{SPD}(n)$, and permutation-quotiented point clouds.

## 1. Introduction

Diffusion models have become a powerful and widely used framework for generative modeling. They define a forward process that gradually corrupts data with noise, and learn a reverse process that transforms noise back into samples, enabling high-quality generation across many domains (Ho et al., 2020; Song et al., 2021). However, a growing number of applications involve data that do not naturally live in $\mathbb{R}^n$: typical examples include spheres, special orthogonal groups $SO(n)$ and spaces of symmetric positive definite (SPD) matrices. Treating such objects in Euclidean space via some embeddings or projections can break intrin-sic constraints, distort distances and volumes, and create coordinate-dependent artifacts. In contrast, modeling directly on the underlying manifold respects the geometry by construction, aligns the noise process with geodesic structure, and yields a principled inductive bias that can improve both sample quality and generalization.

On a Riemannian manifold $(\mathcal{M}, g)$, a canonical drift-free forward process is Riemannian Brownian motion, whose time-marginal density evolves according to the heat equation $\partial_t p_t = \Delta^{\mathcal{M}} p_t$. Following the drift-free formulation of Riemannian score-based generative modeling of De Bortoli et al. (2022), training and sampling require access to the manifold heat kernel (the transition density of Brownian motion), in particular to sample from it and evaluate its log-density/gradients. While several approximations have been proposed—most notably short-time asymptotics and spectral expansions—these can be accurate only in restricted regimes or rely on strong structural assumptions, limiting their practical scope (Lou et al., 2023).

Instead, we adopt a simple viewpoint: *the heat kernel is the solution of a known partial differential equation (PDE) with a known initial condition*. For each fixed $x_0$, it satisfies

$$\partial_t p(t, x) = \Delta^{\mathcal{M}}_x p(t, x), \qquad p(t, \cdot) \to \delta_{x_0} \text{ as } t \downarrow 0. \quad (1)$$

We approximate this object by learning a numerical surrogate of the heat equation. Concretely, we (i) express $\Delta_{\mathcal{M}}$ in explicit coordinates obtained from an embedding or quotient representation, (ii) use a principled short-time asymptotic to provide an approximate initializer at a small $t_0 > 0$, and (iii) train a physics-informed neural network (PINN) to minimize the PDE residual and match the initializer (Raissi et al., 2019; Wang et al., 2023). The resulting surrogate provides both heat-kernel evaluation (and its gradient) and supports heat-kernel-based sampling, enabling drift-free Riemannian diffusion on manifolds beyond the few simple ones.

We evaluate our method across diverse geometries and applications: spherical climate-event density estimation on $S^2$ (volcanoes, earthquakes, floods, wildfires), synthetic mixtures on $SO(3)$, conditional traffic-flow generation on $\mathrm{SPD}(10)$ from the NYC taxi dataset, and class-conditional EEG brain-connectivity generation on $\mathrm{SPD}(n)$. Moreover, for molecule generation, we go beyond the standard $\mathbb{R}^{k \times n}$

---

[1]Korea Advanced Institute of Science and Technology, Korea. Correspondence to: Gyeonghoon Ko <kog@kaist.ac.kr>, Juho Lee <juholee@kaist.ac.kr>.

*Proceedings of the 43rd International Conference on Machine Learning*, Seoul, South Korea. PMLR 306, 2026. Copyright 2026 by the author(s).

point-cloud view and formulate the task on the quotient manifold $\mathbb{R}^{k \times n}/S_n$, explicitly modding out atom permutations. This leads to a novel diffusion process that evolves over permutation-equivalence classes, so atom identities are shared across molecules. Overall, the results show that PDE-based heat-kernel approximation enables practical drift-free Riemannian diffusion beyond simple geometries.

## 2. Related Work

**Riemannian diffusion models.** Riemannian diffusion models generalize Euclidean diffusion/score-based generative modeling to data on a smooth manifold $\mathcal{M}$ by defining a forward diffusion SDE on $\mathcal{M}$ and learning a reverse-time SDE. RSGM (De Bortoli et al., 2022) provides a general formulation and proposes practical training objectives, including Denoising Score Matching (DSM) and Implicit Score Matching (ISM). Complementary likelihood-oriented formulations (Huang et al., 2022) derive a variational framework for an evidence lower bound (ELBO) on manifolds, which requires computing quantities such as the Riemannian divergence. Recent work improves scalability by exploiting additional geometric structure (e.g., symmetric spaces), yielding more accurate and efficient evaluation and approximation of manifold diffusion quantities, including heat-kernel-related terms, in higher dimensions (Lou et al., 2023; Mangoubi et al., 2025). Relatedly, Thornton et al. (2022) propose Riemannian Diffusion Schrödinger Bridge (RDSB), extending diffusion Schrödinger bridges to compact Riemannian manifolds for generative modeling.

**Flow matching on Riemannian manifolds.** Flow matching is an alternative strategy for building generative models on Riemannian manifolds. It trains a continuous-time generative model by learning a time-dependent vector field whose flow transports a simple base distribution to the data distribution, avoiding likelihood computation and divergence estimation. Chen & Lipman (2024) extend conditional flow matching to general geometries via Riemannian Flow Matching (RFM), where the training targets are constructed by geodesic interpolation. This formulation requires only exp and log maps, yielding a scalable and simulation-free objective. This framework has been adopted in geometric generative modeling, including $SE(3)$-equivariant protein modeling and torus-valued torsion-angle generation (Yim et al., 2023; Bose et al., 2024; Huguet et al., 2024; Lin et al., 2024).

**Applications of generative modeling on Riemannian manifolds.** Manifold-aware generative models are motivated by geometric constraints and symmetries, especially in scientific and 3D data. Typical examples include distributions on $S^2$ (e.g., geophysical event locations), rotations on $SO(3)$ for 3D orientation modeling, and rigid-body

frames on $SE(3)$ in robotics and protein structure generation (De Bortoli et al., 2022; Leach et al., 2022; Urain et al., 2023; Yim et al., 2023). In molecular and materials modeling, periodic degrees of freedom such as internal angles and torsions live on products of circles (tori), motivating manifold generative modeling on tori for conformer generation and related tasks (Jing et al., 2022; Miller et al., 2024). Structured matrix manifolds such as $\mathrm{SPD}(n)$ arise in domains including neuroimaging and traffic analytics, motivating diffusion-style generative modeling directly in SPD geometry (Li et al., 2024; Collas et al., 2025; Chen et al., 2025). These applications motivate scalable manifold generative methods that remain applicable when heat kernels are not analytically tractable.

## 3. Background

**Definitions and notations.** Let $\mathcal{M}$ be a complete, connected and boundaryless smooth Riemannian manifold equipped with a Riemannian metric $g$. The metric $g$ is a smooth rank-2 tensor bundle; for each $x \in \mathcal{M}$, the metric $g_x : T_x\mathcal{M} \times T_x\mathcal{M} \to \mathbb{R}$ is a positive definite bilinear function. In a local coordinate chart $(x^1, \ldots, x^n)$, the metric is represented by the matrix $g = (g_{ij})$, and we write $|g| := \det(g_{ij})$. At $x \in \mathcal{M}$, the exponential map $\exp_x : T_x\mathcal{M} \to \mathcal{M}$ is defined by transporting the tangent vector in $T_x\mathcal{M}$ along the geodesic starting from $x$. The inverse of the exponential map $\log_x = \exp_x^{-1}$ is also defined, although it may not be defined globally. A vector field on $\mathcal{M}$ can be written in coordinates with the local frame $\{\partial_i\}_{i=1,\cdots,n}$. Throughout the paper, we use the Einstein summation convention without raising or lowering indices, i.e. the indices appearing twice in the tensor expression are regarded as being summed, e.g., a standard matrix multiplication is written as $(AB)_{ik} = A_{ij}B_{jk}$.

The metric $g$ induces the gradient $\nabla f$ of a smooth function $f : \mathcal{M} \to \mathbb{R}$ via $g(\nabla f, V) = df(V)$ for all vector fields $V$. The divergence of a vector field $V = V_i\partial_i$ is given in local coordinates by $\mathrm{div}(V) = \frac{1}{\sqrt{|g|}} \partial_i(\sqrt{|g|}\, V_i)$, or identically, for the Levi-Civita connection $\nabla$, $\mathrm{div}(V) = \mathrm{tr}(\nabla V)$. The Laplace–Beltrami operator is defined by

$$\Delta^{\mathcal{M}} f = \mathrm{div}(\nabla f) = \frac{1}{\sqrt{|g|}} \partial_i\Big(\sqrt{|g|}\, g_{ij}^{-1} \partial_j f\Big). \quad (2)$$

The (Riemannian) Brownian motion $\mathbf{B}_t^{\mathcal{M}}$ is the diffusion on $\mathcal{M}$ whose infinitesimal generator is $\frac{1}{2}\Delta^{\mathcal{M}}$. More detailed background on Riemannian geometry can be found in, e.g., Lee (2018); Gallot et al. (1990), and for stochastic processes on Riemannian manifolds, see, e.g., Hsu (2002); Elworthy (1998); De Bortoli et al. (2022).

**Riemannian diffusion models.** Let $p_0$ be a distribution on $\mathcal{M}$. We consider the drift-free Riemannian diffusion

$$dX_t = \sqrt{2}d\mathbf{B}_t^{\mathcal{M}}, \qquad X_0 \sim p_0. \qquad (3)$$

A time-dependent scalar noise schedule $dX_t = \sigma_t\, d\mathbf{B}_t^{\mathcal{M}}$ can be introduced via a time reparameterization, so we present this formulation for simplicity. Let $p_t$ be the marginal density of $(X_t)$ with respect to the Riemannian volume measure of $\mathcal{M}$, and $p_{t|0}(x_t|x_0)$ be the transition density (or the heat kernel, since the SDE is simple Brownian). The heat kernel $p_{t|0}$ satisfies the Fokker-Planck equation (or the heat equation)

$$\begin{aligned}
\partial_t p_{t|0}(x \mid x_0) &= \Delta_x^{\mathcal{M}} p_{t|0}(x \mid x_0), \\
\lim_{t \to 0} p_{t|0}(\cdot \mid x_0) &= \delta_{x_0}(\cdot).
\end{aligned} \qquad (4)$$

Under mild regularity assumptions, the reverse-time process $(X_{T-t})_{t \in [0,T]}$ satisfies the reverse-time SDE (De Bortoli et al., 2022)

$$\begin{aligned}
dX_{T-t} &= 2s_{T-t}(X_{T-t})\, dt + \sqrt{2}d\widetilde{\mathbf{B}}_t^{\mathcal{M}}, \\
X_T &\sim p_T,
\end{aligned} \qquad (5)$$

where $\widetilde{\mathbf{B}}_t^{\mathcal{M}}$ is a (reverse-time) Brownian motion on $\mathcal{M}$ and

$$s_t(x) := \nabla_x \log p_t(x) \qquad (6)$$

is the *score function*.

Given a neural network $s_\theta : [0,T] \times \mathcal{M} \to T\mathcal{M}$ with $s_\theta(t,x) \in T_x\mathcal{M}$, one can approximate the score function $s$ by $s_\theta$ by learning $\theta$ with *denoising score matching (DSM)* objective (Vincent, 2011; De Bortoli et al., 2022),

$$\begin{aligned}
&\mathcal{L}_{\mathrm{DSM}}(\theta) := \\
&\mathbb{E}\left[\lambda(t)\left\| s_\theta(t, X_t) - \nabla_{X_t} \log p_{t|0}(X_t \mid X_0) \right\|_{g_{X_t}}^2\right].
\end{aligned} \qquad (7)$$

When $\mathcal{M}$ is Euclidean, this formulation is identical to the standard Euclidean diffusion models (Ho et al., 2020; Song et al., 2021). In this case, the heat kernel $p_{t|0}$ becomes the isotropic Gaussian, so both sampling the forward diffusion $p_{t|0}(\cdot, x_0)$ and computing the conditional score function $\nabla_{x_t} \log p_{t|0}(x_t \mid x_0)$ are tractable. However, when working with a general Riemannian manifold $\mathcal{M}$, the heat kernel is not known in closed form in general. Previous works (De Bortoli et al., 2022; Lou et al., 2023) introduce a few tricks to overcome this challenge.

**Asymptotic methods.** When $t$ is sufficiently small, $p_{t|0}$ can be approximated by a Gaussian-like form:

$$\begin{aligned}
&p_{t|0}(x_t|x_0) \approx G_t(x_t, x_0) \\
&\quad := \frac{1}{(4\pi t)^{n/2}} \exp\left(-\frac{d(x_t, x_0)^2}{4t}\right)
\end{aligned} \qquad (8)$$

where $n = \dim \mathcal{M}$ and $d(\cdot, \cdot)$ is the geodesic distance. This justifies sampling from $p_{t|0}(\cdot|x_0)$ by sampling a Gaussian in a tangent space $T_{x_0}\mathcal{M}$ and push-forwarding by the exponential map $\exp_{x_0} : T_{x_0}\mathcal{M} \to \mathcal{M}$. This distribution is sometimes called the *warped Gaussian*. When $t$ is large, sampling from $p_{t|0}(\cdot|x_0)$ can be approximated by the *geodesic random walk* algorithm (De Bortoli et al., 2022), which is a variant of the random walk algorithm where the Gaussian is replaced by the warped Gaussian. The score function $\nabla \log p_{t|0}$ also admits the asymptotic form:

$$\begin{aligned}
&\nabla_{x_t} \log p_{t|0}(x_t|x_0) \\
&\approx \nabla_{x_t} \log G_t(x_t, x_0) = \frac{1}{2t} \log_{x_t}(x_0),
\end{aligned} \qquad (9)$$

called the Varadhan's asymptotics (Bismut, 1984). However, this approximation does not incorporate the volume element distorted by the curvature of the manifold. To account for this, Lou et al. (2023); Camporesi (1990) introduced the Schwinger-DeWitt approximation:

$$\begin{aligned}
&p_{t|0}(x_t|x_0) \approx G_t^{\mathrm{SD}}(x_t, x_0) \\
&:= \frac{D_{x_0}(x_t)^{-1/2}}{(4\pi t)^{n/2}} \exp\left(-\frac{d(x_t, x_0)^2}{4t} + \frac{tR}{6}\right)
\end{aligned} \qquad (10)$$

where $D_{x_0}(x_t) = |\det(d(\exp_{x_0})_{\log_{x_0}(x_t)})|$ measures the volume distorted by the exponential map, and $R$ is the scalar curvature, assuming $\mathcal{M}$ has a constant scalar curvature. However, this also only works with sufficiently small time $t$.

**Spectral methods.** When the manifold $\mathcal{M}$ is compact, the Laplace-Beltrami operator $\Delta^{\mathcal{M}}$ is a self-adjoint operator on the Hilbert space $L^2(\mathcal{M})$ (with respect to the Riemannian volume measure). Hence it admits a spectral decomposition $\{(\lambda_i, \phi_i)\}_{i=0}^\infty$ satisfying $\Delta^{\mathcal{M}}\phi_i = -\lambda_i\phi_i$, with eigenvalues $0 = \lambda_0 < \lambda_1 \leq \cdots$. The heat kernel then has the spectral expansion

$$p_{t|0}(x \mid x_0) = \sum_{i=0}^{\infty} e^{-\lambda_i t}\, \phi_i(x)\, \phi_i(x_0), \qquad (11)$$

which can be approximated by truncating the series. In practice, however, this only works with *compact* manifolds and requires access to the eigenfunctions of $\Delta^{\mathcal{M}}$, which are rarely available beyond a few highly symmetric examples. Recent work (Lou et al., 2023) stabilizes and accelerates such computations by reformulating the expansion using representation theory on Lie groups, enabling efficient evaluation on certain groups and homogeneous/quotient spaces. Nevertheless, these methods rely on strong algebraic structure and are only applicable to (quotients of) compact Lie groups or Euclidean groups.

**Mapping into Euclidean spaces.** When the exact computation on manifolds is challenging, using their Euclidean counterparts can be a useful trick. De Bortoli et al. (2022) used stereographic projection to map data on 2D sphere into $\mathbb{R}^2$. Li et al. (2024) mimicked the logic of Euclidean diffusion on the space of positive definite matrices, using operations defined on its tangent space. Similar tricks are common for Lie groups: one can perform diffusion updates in the associated Lie algebra (a vector space) via the log and exp maps, e.g., for $SO(3)$ or $SE(3)$ (Leach et al., 2022; Jiang et al., 2023).

## 4. Methods

Given an explicit description of a Riemannian manifold $(\mathcal{M}, g)$, we approximate the heat kernel $p_{t|0}(\cdot|x_0)$ by numerically solving the heat equation $\partial_t p = \Delta^{\mathcal{M}} p$ with the initial condition $p_{0|0} = \delta(\cdot)$. For this work, we first locate the manifold $\mathcal{M}$ into an easily-computable coordinate system, in which we derive the heat equation. Then we choose a properly approximated initial condition, since the original initial condition $p_{0|0} = \delta(\cdot)$ cannot be imposed numerically. Finally, we train a PINN that approximates the heat kernel. The PINN is used for both (i) forward sampling of Brownian motion and (ii) computing the conditional score function for the DSM loss.

### 4.1. Choosing a coordinate system

When working with manifolds mathematically, it is common to choose multiple *local coordinate systems* and patch them to cover the entire manifold. However, when working with manifolds numerically, using multiple coordinate systems would be computationally inconvenient. Instead, we choose to work with a *global coordinate system*, by embedding the manifold $\mathcal{M}$ into a larger space $\tilde{\mathcal{M}}$ (called an *ambient manifold*; typically $\mathbb{R}^N$, possibly with a non-Euclidean metric $\tilde{g}$) as a submanifold or quotient manifold.

For $\tilde{\mathcal{M}} = \mathbb{R}^N$, a submanifold $\mathcal{M} \subseteq \mathbb{R}^N$ can be practically defined by

$$\mathcal{M} = \Big\{ x \in \mathbb{R}^N \ \Big| \ f_i(x) = 0 \text{ for } i = 1, \ldots, k,$$
$$f_j(x) > 0 \text{ for } j = k+1, \ldots, k+l \Big\}. \tag{12}$$

for functions $f_1, \cdots, f_{k+l} : \mathbb{R}^N \to \mathbb{R}$. The manifold inherits the metric $\tilde{g}$ of $\mathbb{R}^N$. For $x \in \mathcal{M}$, the projection map $P(x) : T_x\mathbb{R}^N \to T_x\mathcal{M} \subseteq T_x\mathbb{R}^N$ is given by

$$P(x) = I - \tilde{g}_x^{-1} J(x)^T \left( J(x) \tilde{g}_x^{-1} J(x)^T \right)^{-1} J(x), \tag{13}$$

where $\tilde{g}_x$ is the matrix representation of the metric at $x$, and $J(x) = \big(J_{ij}(x)\big) := \big(\partial_j f_i(x)\big) \in \mathbb{R}^{k \times N}$ is the Jacobian. The ambient metric $\tilde{g}$ and the projection matrix $P(x)$ are suf-

ficient to express various differential-geometric operations on $\mathcal{M}$ in coordinates.

A quotient manifold $\mathcal{M} = \tilde{\mathcal{M}}/G$ is obtained by identifying points related by an isometry group action of $G$. For simplicity, we restrict to a finite or discrete group $G$, so that $\mathcal{M}$ inherits a smooth structure from $\tilde{\mathcal{M}}$; numerically, quotienting can be implemented by treating all points in an orbit as the same state, e.g., via a canonical representative. We do not explicitly address freeness or properness of the group action; although non-free orbits may introduce lower-dimensional singularities in the quotient, these are negligible for the continuous distributions considered in our deep-learning applications.

### 4.2. Fokker–Planck equation in coordinates

Using the coordinates on the ambient manifold $\tilde{\mathcal{M}}$ with the projection matrix $P(x)$, we can express all the required differential operators in coordinates. Let $\tilde{\nabla}$ denote the Levi–Civita connection of $(\tilde{\mathcal{M}}, \tilde{g})$. Although we present the full expressions for generality, in practice, some terms (e.g., the Levi-Civita connection $\tilde{\nabla}$) simplify by choosing a simple ambient manifold $\tilde{\mathcal{M}}$; for instance, when $\tilde{\mathcal{M}}$ is a Euclidean space, $\tilde{\nabla}$ reduces to partial derivatives. The Fokker-Planck equation (or the heat equation) on $\mathcal{M}$ can be explicitly written in coordinates, with expressions depending only on the ambient metric $\tilde{g}$ and the projection matrix $P$:

$$\partial_t p_{t|0}(x|x_0) = \Delta^{\mathcal{M}} p_{t|0}(x|x_0)$$
$$= P_{ij}(x) \tilde{\nabla}_k \Big( P_{j\ell}(x) \, \tilde{g}_{\ell m}^{-1}(x) \, \partial_m \bar{p}_{t|0}(x|x_0) \Big) P_{ki}(x). \tag{14}$$

where $\bar{p}_{t|0}$ denotes any smooth extension of $p_{t|0}$ on $\tilde{\mathcal{M}}$. The detailed derivation is in Section A.1.

**Example: a 2D unit sphere.** Let $S^2 = \{x \in \mathbb{R}^3 \mid \|x\| = 1\}$ be the unit sphere. Although it's standard to use a local coordinate system such as the spherical coordinates, here we directly embed the 2D sphere in the Euclidean manifold $\mathbb{R}^3$. For $x \in S^2$, the projection matrix $P(x) : T_x\mathbb{R}^3 \to T_xS^2 \subseteq \mathbb{R}^3$ is given as

$$P(x) = I - xx^T; \quad P_{ij} = \delta_{ij} - x_i x_j \tag{15}$$

where $\delta_{ij}$ is the Kronecker delta. In the ambient manifold $\mathbb{R}^3$, the metric $\tilde{g}$ is the identity matrix, and the Levi-Civita connection $\tilde{\nabla}_i$ simplifies to the partials $\partial_i$. Now we can work out the Laplace-Beltrami operator:

$$\Delta^{S^2} f = P_{ij}\partial_i\partial_j f + P_{ki}(\partial_k P_{ij})\partial_j f$$
$$= (\delta_{ij} - x_i x_j)\partial_i\partial_j f - 2x_i\partial_i f. \tag{16}$$

### 4.3. Initial condition for the Fokker–Planck equation

Since it is not possible to impose the initial condition $p_{0|0}(x \mid x_0) = \delta_{x_0}(x)$ directly, we instead initialize the

PDE using the small-time asymptotics of the heat kernel. Let $B_{x_0}(r_{\max}) = \{x \in \mathcal{M} \mid d(x, x_0) \leq r_{\max}\}$ be the geodesic ball of radius $r_{\max}$ centered at $x_0$. On $x \in B_{x_0}(r_{\max})$, the heat kernel admits the zeroth-order expansion

$$
\begin{aligned}
p_t(x \mid x_0) &= G_t^{(0)}(x, x_0) + O(t), \\
G_t^{(0)}(x, x_0) &:= D_{x_0}(x)^{-1/2} \, G_t(x, x_0) \\
&= \frac{D_{x_0}(x)^{-1/2}}{(4\pi t)^{n/2}} \exp\left(-\frac{d(x, x_0)^2}{4t}\right).
\end{aligned}
\tag{17}
$$

A principled way to obtain tighter approximations is the parametrix method (also known as the Minakshisundaram–Pleijel recursion) (Rosenberg, 1997), which yields an order-$n$ expansion of the form

$$
\begin{aligned}
p_t(x|x_0) &= G_t^{(n)}(x, x_0) + O(t^n), \\
G_t^{(n)}(x, x_0) &:= G_t(x, x_0)\Big(u_0(x, x_0) \\
&\quad + u_1(x, x_0)t + \cdots + u_{n-1}(x, x_0)t^{n-1}\Big).
\end{aligned}
\tag{18}
$$

Here $u_0(x, x_0) = D_{x_0}(x)^{-1/2}$, and $u_1, \ldots, u_{n-1}$ can be computed recursively; we defer the details to Section A.2. The Schwinger-DeWitt approximation Equation (10) is also a simplified version of the first-order parametrix expansion.

**Stability analysis of PDE.** In general, similarity in initial condition for PDEs does not guarantee similarity of the corresponding solutions at later times. However, due to various stability properties of the heat equation, solving the heat equation from an approximate initial condition at $t = t_0$ yields a solution that remains reliable for all $t \geq t_0$.

**Theorem 4.1.** *Let $\tilde{p}_t(x) : [t_0, \infty) \times \mathcal{M} \to \mathbb{R}$ be the (unique) solution of the evolution equation $\partial_t \tilde{p}_t(x) = \Delta^{\mathcal{M}} \tilde{p}_t(x)$ with the initial condition $\tilde{p}_{t_0}(x) = p_{t_0|0}(x|x_0) + \varepsilon_{t_0}(x)$ with small initialization error*

$$
|\varepsilon_{t_0}(x)| \leq \varepsilon \, p_{t_0|0}(x|x_0) \quad \forall x \in \mathcal{M}
\tag{19}
$$

*for an error level $\varepsilon \in (0, 1)$. Then, on the domain $[t_0, t_{\max}] \times B_{x_0}(r_{\max})$, the solution $\tilde{p}_{t|0}(x)$ is the approximation of $p_{t|0}(x|x_0)$ with the same order of accuracy as the initial condition:*

$$
\begin{aligned}
\tilde{p}_t(x) &= p_{t|0}(x|x_0) + \varepsilon_t(x), \\
|\varepsilon_t(x)| &\leq \varepsilon \, p_{t|0}(x|x_0).
\end{aligned}
\tag{20}
$$

*Moreover, under mild regularity assumption on $\tilde{p}_{t_0}(x)$ and its gradients, this solution can also approximate the log-density and the score:*

$$
\begin{aligned}
\log \tilde{p}_t(x) &= \log p_{t|0}(x|x_0) + O(\varepsilon) \\
\nabla_x \log \tilde{p}_t(x) &= \nabla_x \log p_{t|0}(x|x_0) + O_\delta(\varepsilon).
\end{aligned}
\tag{21}
$$

*where the last statement holds $\forall \, \delta > 0$, on $t \in [t_0 + \delta, t_{\max}]$.*

Theorem C.1 is a rigorous version of this theorem. Moreover, in Section C.2 we extend this result to learned PINNs with small log-PDE residual, where the log-density error is bounded by the boundary error plus the accumulated residual.

### 4.4. Training PINN

A PINN is a deep-learning approach for approximating solutions to PDEs by embedding the governing equation directly into the training objective (Raissi et al., 2019). While many PINN variants also incorporate observations at interior collocation (quadrature) points, we describe here a basic formulation that uses only the initial/boundary condition and the PDE residual. Let the space–time domain be $[t_0, t_{\max}] \times U$, where $U \subset \mathbb{R}^N$ is closed, and consider a well-posed evolution equation written in residual form

$$
\mathcal{R}[f](t, x) := \mathcal{R}(t, x, \partial_t f, \partial_x f, \partial_x^2 f, \cdots) = 0
\tag{22}
$$

with initial condition $f(t_0, x) = f_0(x)$. In the PINN framework, we parameterize the solution by a neural network $f_\theta : [t_0, t_{\max}] \times U \to \mathbb{R}$ and train $\theta$ minimizing a combination of the initial/boundary loss:

$$
\begin{aligned}
\mathcal{L}_B(f_\theta) &:= \mathbb{E}_x[\|\|f_\theta(x, t_0) - f_0(x)\|\|^2] \\
\mathcal{L}_I(f_\theta) &:= \mathbb{E}_{t,x}[\|\|\mathcal{R}[f_\theta](x, t)\|\|^2]
\end{aligned}
\tag{23}
$$

where $x \in U$ and $(t, x) \in [t_0, t_{\max}] \times U$ are sampled on some adequate distributions on their domain.

This PINN formulation is well suited to approximating the heat kernel. For a given manifold $\mathcal{M}$, we derive the heat equation written in coordinates and train the PINN with the approximate initial condition $\tilde{p}_{t_0|0}$. For numerical stability, we let $p_{t|0} = e^{\phi_{t|0}}$ and rewrite the equation in terms of $\phi_{t|0}$, and train a PINN $\phi_{t|0}^{(\theta)}$ approximating it. The trained PINN is used jointly with the short-time approximation, to approximate the heat kernel:

$$
\log p_{t|0}^{(\text{approx})}(x|x_0) = \begin{cases} \log \tilde{p}_{t|0}(x|x_0), \quad t < t_0, \\ \text{(short-time asymptotics)} \\ \phi_{t|0}^{(\theta)}(x|x_0), \quad t \geq t_0. \\ \text{(PINN approximation)} \end{cases}
\tag{24}
$$

For stable training of PINN, we adopt several techniques of Wang et al. (2023), including modified MLP structures with embeddings, adaptive weight scaling of two losses, and time-scheduled learning method.

**Representational power of PINN.** Quantifying whether a PINN can accurately represent the heat kernel on a given (potentially large and geometrically complex) manifold $\mathcal{M}$ is challenging, since the approximation error depends strongly on modeling choices such as the network architecture. Nevertheless, the heat equation has a smoothing property: for

any $t > t_0$, the evolved log-density $\log p_{t|0}$ is guaranteed to be more regular than the initial log-density $\log p_{t_0|0}$, e.g., in terms of Sobolev norms (Grigor'yan, 2012). This suggests that, once a reasonable approximation is available at $t = t_0$, learning the forward solution for $t \geq t_0$ may be easier than fitting the initial condition itself. In high dimensions, PINNs can be difficult to train and may require more computational and memory resources (Cho et al., 2023; Hu et al., 2024), often necessitating additional inductive biases (e.g., incorporating known symmetries and using expressive input embeddings) to improve optimization behavior.

### 4.5. Riemannian diffusion model with PINN

All ingredients for building Riemannian diffusion models are now set. We train a Riemannian diffusion model as described in Section 3. Having knowledge of the density $\log p_{t|0}^{(\text{approx})}(x|x_0)$, the forward sampling $x_t \sim p_{t|0}^{(\text{approx})}(\cdot|x_0)$ can be done by a Markov chain Monte Carlo (MCMC) method. Since the warped Gaussian discussed in Section 3 is an effective approximation of $p_{t|0}$, we use it for the initial distribution of the MCMC, then run a geodesic random walk for a fixed step size $\varepsilon > 0$ for proposal. After $x_t$ is drawn, the conditional score $\nabla \log p_{t|0}^{(\text{approx})}$ is computed by differentiating the PINN, and train the diffusion model under the DSM loss.

The architecture of a diffusion model on $\mathcal{M}$ depends on the geometry of $\mathcal{M}$. For example, if $\mathcal{M}$ is (a quotient of) a submanifold of an ambient space $\tilde{\mathcal{M}} = \mathbb{R}^N$, then an extrinsic network $s_\theta : [0, T] \times \mathbb{R}^N \to \mathbb{R}^N$ can be used, possibly with appropriate symmetry constraints induced by the quotient. Alternatively, one may use an intrinsic architecture tailored to $\mathcal{M}$, e.g., $s_\theta : [0, T] \times \mathcal{M} \to \mathcal{M}$. In this case, since the denoiser corresponds to a tangent vector $\nabla_x \log p_{t|0}(x \mid x_0)$, we adopt a standard trick from Euclidean diffusion models that reparameterizes prediction in terms of the denoised data (Ho et al., 2020). Concretely, instead of directly learning $\nabla_x \log p_{t|0}(x \mid x_0)$, we train the "denoised input" $\exp_x\big(2t\nabla_x \log p_{t|0}(x \mid x_0)\big) \in \mathcal{M}$.

For the reverse sampling, we also require the perturbed (or prior) distribution $p_T \approx p(\cdot|x_0)$. When $\mathcal{M}$ is compact, we use the uniform distribution $p_T \approx \text{Unif}(\mathcal{M})$ assuming $T$ is sufficiently large. When $\mathcal{M}$ is not compact, we set a rough mean of the data distribution $x_{\text{mean}}$ and take $p_T \approx p_T(\cdot, x_{\text{mean}})$. This setting is common in recent state-of-the-art Euclidean diffusion models (Karras et al., 2022).

## 5. Worked examples

### 5.1. 2D sphere $S^2$

As derived in Section 4.2, the heat equation on $S^2 \subset \mathbb{R}^3$ is given as $\partial_t p = (\delta_{ij} - x_i x_j)\partial_i \partial_j p - 2x_i \partial_i p$, and for

$\phi = \log p$,

$$
\begin{aligned}
\partial_t \phi &= [(\delta_{ij} - x_i x_j)(\partial_i \partial_j \phi + \partial_i \phi \partial_j \phi) - 2x_i \partial_i \phi] \\
&= \text{tr}(\nabla^2 \phi) + \|\nabla \phi\|^2 - x^T(\nabla^2 \phi)x \\
&\quad - (x^T \nabla \phi)^2 - 2x^T \nabla \phi
\end{aligned}
\tag{25}
$$

The initial condition can be computed using the parametrix method as described in Section 4.3 up to third order. The recursion integrals are evaluated via series expansion using a computer algebra system (CAS). Detailed procedure is in Section A.3.

### 5.2. Special orthogonal group $SO(3)$

The special orthogonal group $SO(3)$ is isometric to $S^3/\{\pm I\}$, where $S^3 \subset \mathbb{R}^4$ is the 3D unit sphere and $\cdot/\{\pm I\}$ denotes quotienting by identifying the opposite points on sphere, i.e. $\pm x \in S^3$ are regarded as an identical point. This isometry can be explicitly written using quaternions. Once the mapping is constructed, the heat equation is similar to the case of the 2D sphere:

$$
\begin{aligned}
\partial_t \phi &= [(\delta_{ij} - x_i x_j)(\partial_i \partial_j \phi + \partial_i \phi \partial_j \phi) - 3x_i \partial_i \phi] \\
&= \text{tr}(\nabla^2 \phi) + \|\nabla \phi\|^2 - x^T(\nabla^2 \phi)x \\
&\quad - (x^T \nabla \phi)^2 - 3x^T \nabla \phi
\end{aligned}
\tag{26}
$$

and the initial condition is again computed using the parametrix method, similar to Section 5.1.

### 5.3. Space of positive-definite matrices $\text{SPD}(n)$

Generative modeling on the space of positive definite matrices $\text{SPD}(n) = \{X \in \mathbb{R}^{n \times n} \mid X \text{ is positive definite}\}$ has drawn interest in recent research, e.g. for traffic analysis or fMRI data modeling Li et al. (2024); Collas et al. (2025). The hyperbolic geometry of $\text{SPD}(n)$ is well suited to the affine-invariant metric $g_X(U, V) = \text{tr}(X^{-1}UX^{-1}V)$, written in infinitesimal form. The derivation of the heat equation is in Section A.4. For any initial point $X_0 \in \text{SPD}(n)$, since the map $X \mapsto X_0^{-1/2}XX_0^{-1/2}$ is an isometry under the affine-invariant metric that maps $X_0$ to $I \in \text{SPD}(n)$, it suffices to consider the heat equation starting at $I$. The heat equation starting at $I$ only depends on the eigenvalues of $X \in \text{SPD}(n)$, so we parametrize the matrices by their log-eigenvalues $(r_1, \cdots, r_n)$ where $X = Q \, \text{diag}(e^{r_1}, \cdots, e^{r_n})Q^T$ for some $Q \in O(n)$, then the heat kernel is reformulated by an equation in $r_1, \cdots, r_n$:

$$
\partial_t p_{t|0} = \sum_i \partial_i^2 p_{t|0} + \frac{1}{2}\sum_{i<j} \coth\left(\frac{r_i - r_j}{2}\right)(\partial_i p_{t|0} - \partial_j p_{t|0})
\tag{27}
$$

where the partials are in the $r$-coordinates, i.e. $\partial_i = \frac{\partial}{\partial r_i}$. For the initial condition, we use the 0-th order approximation in Section 4.3:

$$
\tilde{p}_{t_0|0}(X|I) = \frac{D(r)^{-1/2}}{(4\pi t_0)^{n(n+1)/4}} \exp\left(-\frac{\|r\|^2}{4t_0}\right)
\tag{28}
$$

where $D(r) = \prod_{i<j} \frac{\sinh((r_i - r_j)/2)}{(r_i - r_j)/2}$.

## 5.4. Space of point clouds quotiented by permutation symmetry $\mathbb{R}^{k \times n}/S_n$

A point cloud $X = [x^{(1)}, \cdots, x^{(n)}] \in \mathbb{R}^{k \times n}$ usually has the permutation symmetry $S_n$ by reordering the points, and it is common to incorporate permutation symmetry for e.g. architectural designs (Qi et al., 2017; Zaheer et al., 2017; Lee et al., 2019) or probabilistic formulation (Garnelo et al., 2018) in deep learning. Although existing works for generative modeling of point clouds (e.g., molecules) mostly use permutation symmetric neural networks for denoisers (Hoogeboom et al., 2022; Le et al., 2024), their diffusion process is formulated in $\mathbb{R}^{k \times n}$. To model a diffusion on the quotient manifold $\mathbb{R}^{k \times n}/S_n$, one requires a heat kernel on this space, given as:

$$
\begin{aligned}
p_{t|0}(x \mid x_0) = &\sum_{\sigma \in S_n} \frac{1}{(4\pi t)^{kn/2}} \\
&\cdot \exp\left( -\frac{1}{4t} \sum_{i=1}^{n} \|x^{(i)} - x_0^{(\sigma(i))}\|^2 \right).
\end{aligned}
\tag{29}
$$

Intuitively, in this setting, all the points $x^{(1)}, \cdots, x^{(n)}$ share the identity. Although the heat kernel is in closed-form, it is intractable to compute due to the $n!$ elements of $S_n$. We use a permutation-symmetric neural network as a PINN to approximate this kernel, with the equation and (approximate) initial condition same as the Euclidean one:

$$
\partial_t \tilde{p}_{t|0} = \sum_{i=1}^{n} \sum_{j=1}^{k} \frac{\partial^2}{\partial x_j^{(i)}}^2 \tilde{p}_{t|0},
$$

$$
\tilde{p}_{t_0|0}(x \mid x_0) = \frac{1}{(4\pi t_0)^{\frac{kn}{2}}} \exp\left( -\frac{\sum_{i=1}^{n} \|x^{(i)} - x_0^{(i)}\|^2}{4t_0} \right).
\tag{30}
$$

## 6. Experiments

In this section, we demonstrate how our method applies across diverse settings and a range of manifolds. Our experiments are designed to highlight the generality of the framework, while minimizing task-specific engineering aimed solely at boosting performance. Our implementation is available at https://github.com/kogyeonghoon/riem-diff-pinn.git.

### 6.1. Training PINN

We largely follow the PINN training protocol of Wang et al. (2023), adapting its architectural and optimization strategies to our manifold setting. Specifically, we employ a modified MLP architecture in which each hidden layer is fused with manifold-aware coordinate embeddings. This design enriches the coordinate-based neural representation and improves its ability to approximate nonlinear and geometrically complex surrogate solutions. Since the accuracy of later-time dynamics depends critically on the quality of the solution learned at earlier times, we adopt a curriculum strategy that progressively expands the training interval to $[t_0, t_{\max}]$. In addition, we use an adaptive loss-balancing scheme based on the exponential moving average of gradient norms, together with time-dependent residual weights that encourage the PINN to learn the solution in a temporally causal manner.

The input embeddings are chosen according to the geometry of each manifold. For $S^2$ and $SO(3)$, we use Fourier embeddings following Wang et al. (2023). For SPD($n$), we use the symmetric and antisymmetric embeddings described in Section B, motivated by the permutation-invariant structure of the radial heat kernel. For $\mathbb{R}^{k \times n}/S_n$, we replace the modified MLP with a permutation-symmetric GNN architecture, reflecting the permutation symmetry of the quotient heat-kernel surrogate. The spatial and temporal training domains of the PINN, specified by the radius cutoff $r_{\max}$ and the time interval $[t_0, t_{\max}]$, are selected separately for each manifold and downstream dataset. We choose $r_{\max}$ to cover the region where the heat-kernel surrogate is queried, while $t_{\max}$ is set large enough so that the forward heat diffusion sufficiently corrupts the data by the terminal time.

To directly validate the learned heat-kernel surrogate, we evaluate it separately from the downstream diffusion-model performance. On $S^2$ and $SO(3)$, where accurate reference heat-kernel approximations are available via Lou et al. (2023), we compare both the learned log-density $\log p_{t|0}^{(\theta)}(x|x_0)$ and the conditional score $\nabla_x \log p_{t|0}^{(\theta)}(x|x_0)$ against the reference quantities across several noise levels in Table 1. For the remaining manifolds, such reference kernels are not tractable, so we instead assess whether the trained PINN satisfies the defining heat-kernel problem. Specifically, for all manifolds we report the normalized initial/boundary-condition error and the normalized PDE residual of the learned log-density in Table 2. The PDE residual is normalized by the largest-magnitude term appearing in the equation; for example, for a PDE written in the form $A = B + C$, we report $|A - B - C| / \max|A|, |B|, |C|$. These diagnostics measure, respectively, how well the surrogate matches the prescribed short-time initial condition and how accurately it satisfies the manifold heat equation after training. Thus, the quality of the surrogate is evaluated both by direct comparison to existing heat-kernel approximations when available, and by geometry-agnostic PINN consistency checks across all considered manifolds. Finally, we report the wall-clock runtime of the surrogate in Section B.6; since the surrogate is implemented in JAX and evaluated after JIT compilation, log-density and score evaluations incur only a small computational overhead.

*Table 1.* Direct validation of the learned heat-kernel surrogate on $S^2$ and $SO(3)$ against the reference approximation of Lou et al. (2023). We report absolute and relative errors for both the log-density and the conditional score. Large relative score errors at large $t$ are due to the small reference score norm $\|\nabla \log p\|$.

| $\mathcal{M}$ | $t$ | $|\operatorname{err}(\log p)|$ | $\frac{|\operatorname{err}(\log p)|}{|\log p|}$ | $\|\operatorname{err}(\nabla \log p)\|$ | $\frac{\|\operatorname{err}(\nabla \log p)\|}{\|\nabla \log p\|}$ |
|---|---|---|---|---|---|
| $S^2$ | 0.30 | 0.0054 | 0.0009 | 0.0281 | 0.0247 |
| | 0.50 | 0.0079 | 0.0032 | 0.0182 | 0.1844 |
| | 1.00 | 0.0016 | 0.0007 | 0.0027 | 0.0894 |
| | 2.00 | 0.0016 | 0.0006 | 0.0006 | 0.1211 |
| | 4.00 | 0.0016 | 0.0006 | 0.0002 | 2.2570 |
| $SO(3)$ | 0.30 | 0.0006 | 0.0002 | 0.0024 | 0.0277 |
| | 0.50 | 0.0055 | 0.0013 | 0.0060 | 0.2112 |
| | 1.00 | 0.0029 | 0.0007 | 0.0004 | 0.7983 |
| | 2.00 | 0.0058 | 0.0014 | 0.0000 | 4.3249 |
| | 4.00 | 0.0232 | 0.0054 | 0.0077 | 4640.7837 |

*Table 2.* PINN error of the learned heat-kernel surrogate. For each manifold, the first row lists the diagnostic: "BC" denotes the normalized initial/boundary-condition error, and each time value denotes the normalized PDE residual evaluated at that time. The second row reports the corresponding normalized error.

| $\mathcal{M}$ | | | Normalized PINN error | | | |
|---|---|---|---|---|---|---|
| $S^2$ | BC | $t=0.30$ | $t=0.50$ | $t=1.00$ | $t=2.00$ | $t=4.00$ |
| | 0.0002 | 0.0037 | 0.0285 | 0.0185 | 0.0181 | 0.0124 |
| $SO(3)$ | BC | $t=0.30$ | $t=0.50$ | $t=1.00$ | $t=2.00$ | $t=4.00$ |
| | 0.0002 | 0.0278 | 0.0769 | 0.0817 | 0.0427 | 0.1239 |
| $SPD(10)$ | BC | $t=0.10$ | $t=0.30$ | $t=0.50$ | $t=1.00$ | – |
| | 0.1490 | 0.0048 | 0.0035 | 0.0034 | 0.0060 | – |
| $\mathbb{R}^{k \times n}/S_n$ | BC | $t=0.01$ | $t=0.03$ | $t=0.10$ | $t=0.20$ | $t=0.40$ |
| | 0.0021 | 0.0066 | 0.1391 | 0.1543 | 0.1941 | 0.1747 |

## 6.2. Climate science datasets on $S^2$

Following the experimental setup of RSGM (De Bortoli et al., 2022), we validate our method on the spherical density-estimation benchmarks comprising geolocated events on Earth's surface—volcanic eruptions (NGDC/WDS, 2022b), earthquakes (NGDC/WDS, 2022a), floods (Brakenridge, 2017), and wildfires (EOSDIS, 2020). Each event is represented as a point on the unit sphere $S^2$. We report test log-likelihoods computed with the same manifold-adapted likelihood estimator for score-based models used in RSGM, and we compare against their method. The results are summarized in Table 3. In RSGM, the computation of the heat kernel is approximated using a combination of the Varadhan approximation and spectral decomposition.

## 6.3. Synthetic data on $SO(3)$

We again follow the experimental setting of RSGM and evaluate our method on $SO(3)$ using synthetic data drawn from a mixture of wrapped Gaussians. In RSGM, the heat kernel is approximated using the Varadhan approximation. Results are reported in Table 3.

*Table 3.* Test log-likelihoods on geolocation benchmarks on $S^2$ and synthetic data on $SO(3)$.

| | Volcano | Earthquakes | Floods | Wildfires | Synthetic ($SO(3)$) |
|---|---|---|---|---|---|
| RSGM | $3.51 \pm 0.12$ | $0.10 \pm 0.03$ | $-0.53 \pm 0.04$ | $1.03 \pm 0.01$ | $0.18 \pm 0.01$ |
| Ours | $\mathbf{3.56 \pm 0.17}$ | $\mathbf{0.24 \pm 0.05}$ | $\mathbf{-0.47 \pm 0.01}$ | $\mathbf{1.08 \pm 0.01}$ | $\mathbf{0.21 \pm 0.003}$ |

## 6.4. Traffic analysis on $SPD(10)$

Li et al. (2024) is one of the early works to formulate diffusion on the space of positive definite matrices. They use the NYC taxi dataset (Tucker et al., 2023), where New York City is divided into 10 regions and each $SPD(10)$ matrix encodes traffic flows between pairs of regions. The dataset includes 13 predictor features, such as weather conditions on the observed day. Following their experimental setting, we train a conditional diffusion model that takes predictors $y$ as input and generates an SPD matrix $X \in SPD(10)$. We evaluate using the Frobenius distance between generated matrices and the ground-truth data. Results are in Table 4.

*Table 4.* Mean and quartiles of Frobenius distances between true and generated data.

| Method | Mean | 25% | 50% | 75% |
|---|---|---|---|---|
| SPD-DDPM | $6.15 \pm 0.12$ | $1.75 \pm 0.05$ | $3.40 \pm 0.05$ | $7.49 \pm 0.20$ |
| Ours | $\mathbf{5.21 \pm 0.12}$ | $\mathbf{1.70 \pm 0.01}$ | $\mathbf{3.12 \pm 0.07}$ | $\mathbf{6.31 \pm 0.36}$ |

## 6.5. Brain connectivity (EEG) analysis on $SPD(n)$

DiffeoCFM (Collas et al., 2025) introduces conditional flow matching using a diffeomorphic mapping $SPD(n) \rightarrow \mathbb{R}^{n(n+1)/2}$, enabling SPD matrices to be modeled in a Euclidean space. Following their experimental protocol, we evaluate our method on the EEG datasets BNCI2014-002 (Steyrl et al., 2016) and BNCI2015-001 (Faller et al., 2012). Each SPD matrix is paired with a binary motor-imagery label, and we generate SPD samples conditionally on label. We evaluate the generated samples using distributional quality metrics ($\alpha$-precision, $\beta$-recall and F1) based on Alaa et al. (2022), and downstream classification accuracy (AUC and F1), obtained by training a classifier on generated samples and evaluating it on the test set.

In addition to DiffeoCFM, we compare against the diffusion-style baseline LowTriDDPM (Collas et al., 2025), which models diffusion over the lower-triangular entries of SPD matrices. As shown in Table 5 and Table 6, DiffeoCFM performs best on downstream AUC/CAS-F1, while our method is competitive and achieves strong precision/Q-F1 on some settings. We hypothesize that the data-centric diffeomorphic parameterization (via a log map around a central reference point) captures global structure more effectively than operating directly on $SPD(n)$ under the affine-invariant metric.

*Table 5.* Quality and CAS metrics on BNCI 2014-002.

| Method | $\alpha$-P | $\beta$-R | Q-F1 | AUC | CAS-F1 |
|---|---|---|---|---|---|
| LowTriDDPM | $0.36 \pm 0.05$ | $\mathbf{0.76 \pm 0.03}$ | $0.47 \pm 0.04$ | $0.65 \pm 0.03$ | $0.66 \pm 0.06$ |
| DiffeoCFM | $0.62 \pm 0.06$ | $0.63 \pm 0.05$ | $0.61 \pm 0.02$ | $\mathbf{0.81 \pm 0.01}$ | $\mathbf{0.74 \pm 0.02}$ |
| Ours | $\mathbf{0.75 \pm 0.04}$ | $0.35 \pm 0.04$ | $\mathbf{0.69 \pm 0.02}$ | $0.75 \pm 0.02$ | $0.70 \pm 0.03$ |

*Table 6.* Quality and CAS metrics on BNCI 2015-001.

| Method | $\alpha$-P | $\beta$-R | Q-F1 | AUC | CAS-F1 |
|---|---|---|---|---|---|
| LowTriDDPM | $0.76 \pm 0.03$ | $\mathbf{0.93 \pm 0.01}$ | $0.84 \pm 0.04$ | $0.64 \pm 0.03$ | $0.56 \pm 0.10$ |
| DiffeoCFM | $\mathbf{0.93 \pm 0.01}$ | $0.86 \pm 0.01$ | $\mathbf{0.90 \pm 0.01}$ | $\mathbf{0.73 \pm 0.01}$ | $\mathbf{0.67 \pm 0.01}$ |
| Ours | $0.91 \pm 0.02$ | $0.81 \pm 0.02$ | $\mathbf{0.90 \pm 0.01}$ | $0.71 \pm 0.01$ | $0.56 \pm 0.01$ |

### 6.6. Molecule generation on $\mathbb{R}^{k \times n}/S_n$

Finally, we study molecule generation on the permutation-quotiented space $\mathbb{R}^{k \times n}/S_n$, using the E(3)-Equivariant Diffusion Model (EDM) (Hoogeboom et al., 2022) as a baseline. QM9 (Ramakrishnan et al., 2014) contains molecules with up to 29 atoms, along with atomic positions and molecular properties. Following EDM, we concatenate atomic coordinates with one-hot atom types to form per-atom feature vectors, representing a molecule with $n$ atoms as a point cloud in $\mathbb{R}^{k \times n}$. To enable a more direct comparison with our method, we train an EDM variant based on the variance-exploding (VE) SDE (Song et al., 2021). We report molecule stability, validity, uniqueness, and novelty in Table 7. We find that our method produces more valid molecules but lower novelty, reflecting a common validity–novelty trade-off in molecule generation. We also note that our method is not compared to the state-of-the-art baselines (Le et al., 2024; Irwin et al., 2025) which use more parameters and incorporate edge features.

## 7. Conclusion

In this work, we addressed a central practical bottleneck in drift-free Riemannian diffusion models: training and sampling typically require access to the manifold heat kernel and its gradients, which are rarely available beyond a few simple manifolds. We proposed a general, PDE-based alternative: given an explicit specification of a manifold, we (i) select a global coordinate representation, (ii) derive the corresponding heat (Fokker–Planck) equation in coordinates, (iii) replace the singular Dirac-delta initial condition with a principled short-time asymptotic initializer, and (iv) train a physics-informed neural network (PINN) to approximate the log heat kernel. The resulting surrogate heat kernel supports both forward noising and conditional-score evaluation, enabling denoising score matching and reverse-time sampling on manifolds where closed-form kernels are unavailable.

Empirically, we demonstrated that this *solve-the-heat-equation* viewpoint can serve as a practical backend for manifold diffusion modeling across diverse geometries and data modalities, including density estimation

*Table 7.* QM9 generation metrics (all in %).

| Method | M-Stab | Val | Uniq | Nov |
|---|---|---|---|---|
| EDM (VE) | $95.20 \pm 0.36$ | $98.13 \pm 0.42$ | $\mathbf{99.12 \pm 0.15}$ | $\mathbf{63.30 \pm 2.41}$ |
| Ours | $\mathbf{98.63 \pm 0.25}$ | $\mathbf{99.60 \pm 0.26}$ | $84.61 \pm 3.11$ | $41.70 \pm 1.59$ |

on $S^2$, synthetic modeling on $SO(3)$, conditional generation on $\mathrm{SPD}(n)$, and quotient-manifold settings such as permutation-invariant point clouds for molecule generation. Overall, the experiments support the thesis that PDE-based heat-kernel approximation can broaden the applicability of Riemannian diffusion models beyond cases where spectral methods or analytic kernels are readily available.

**Limitations.** First, training a PINN for the heat kernel can become nontrivial as the effective dimension grows or when the manifold geometry/topology induces a highly structured solution; in such regimes, optimization can be unstable and may require stronger inductive biases (e.g., symmetry-aware architectures or better feature embeddings). Second, the overall pipeline is computationally heavy in practice: PINN training requires many collocation evaluations with higher-order autodifferentiation, and downstream usage may further require repeated evaluations (and sometimes MCMC-style forward sampling), leading to substantial runtime and engineering overhead. Finally, while the method is general, it is also more complicated than alternatives such as (Riemannian) flow matching, which typically bypass heat-kernel estimation entirely and can be simpler to implement when accurate exponential/log maps are available.

**Future work.** A key direction is to automate and systematize the full workflow so that applying the method to a new manifold becomes closer to "plug-and-play." Concretely, this includes more automatic identification/derivation of the governing PDE (e.g., Laplace–Beltrami in chosen coordinates), robust construction of the short-time initializer and boundary/domain specification, and more reliable PINN training protocols. Beyond improving usability and scalability, it is also promising to extend the PDE-surrogate idea to a broader class of generative modeling frameworks—e.g., manifold Schrödinger bridges and controlled diffusions, or hybrids that combine the geometric convenience of flow matching with stochasticity—where learned transition-density surrogates (or their score analogues) could serve as modular building blocks.

## Acknowledgements

This work was partly supported by Institute of Information & communications Technology Planning & Evaluation(IITP) grant funded by the Korea government(MSIT) (No.RS-2019-II190075, Artificial Intelligence Graduate School Program(KAIST), No.RS-2024-00509279, Global AI Frontier Lab, and No.RS-2022-II220713, Meta-learning Applicable to Real-world Problems).

## Impact Statement

This work advances foundational methods for generative modeling on Riemannian manifolds. We do not anticipate direct societal impact from the results presented. While the techniques could be used in downstream applications, they do not introduce new application domains or materially change the known risk profile of modern generative models; no specific ethical concerns beyond standard responsible-use considerations are expected.

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

# A. Mathematical Details

## A.1. Fokker-Planck equation in coordinates

The notation follows from Section 4.2.

**Gradient, divergence and Laplace-Beltrami.** For a smooth function $f : \mathcal{M} \to \mathbb{R}$, to work in the ambient manifold $\tilde{\mathcal{M}}$, we use any smooth extension $\bar{f}$ to a neighborhood of $\mathcal{M}$ in $\tilde{\mathcal{M}}$. The *gradient* of $\bar{f}$ in the ambient manifold $\tilde{\mathcal{M}}$ is given by

$$(\widetilde{\nabla}\bar{f})_i = \tilde{g}_{ij}^{-1}\, \partial_j \bar{f},$$

and the gradient of $f$ in the submanifold $\mathcal{M}$ is obtained by projection:

$$(\nabla f)(x) = P(x)\,\widetilde{\nabla}\,\bar{f}(x), \qquad (\nabla f)_i = P_{ij}(x)\,\tilde{g}_{jk}^{-1}(x)\,\partial_k \bar{f}(x). \tag{31}$$

For a smooth tangent vector field $X$ on $\mathcal{M}$ (so $PX = X$), the *divergence* is the trace over tangent directions:

$$\operatorname{div} X = \operatorname{tr}\big(P\,(\tilde{\nabla}X)\,P\big) = P_{ij}(x)\,(\tilde{\nabla}_k X)_j\,P_{ki}(x). \tag{32}$$

The *Laplace–Beltrami operator* is $\Delta^{\mathcal{M}} f := \operatorname{div}(\nabla f)$, hence

$$\Delta^{\mathcal{M}} f = \operatorname{tr}\Big(P\,\tilde{\nabla}\big(P\,\widetilde{\nabla}\,\bar{f}\big)\,P\Big) = P_{ij}(x)\,\tilde{\nabla}_k\Big(P_{j\ell}(x)\,\tilde{g}_{\ell m}^{-1}(x)\,\partial_m \bar{f}(x)\Big)\,P_{ki}(x). \tag{33}$$

**Fokker–Planck equation.** Therefore, the Fokker–Planck equation of the simple Brownian motion is given by

$$\partial_t p_{t|0}(x \mid x_0) = \Delta^{\mathcal{M}} p_{t|0}(x \mid x_0) = P_{ij}(x)\,\tilde{\nabla}_k\Big(P_{j\ell}(x)\,\tilde{g}_{\ell m}^{-1}(x)\,\partial_m \bar{p}_{t|0}(x \mid x_0)\Big)\,P_{ki}(x). \tag{34}$$

Note that, although $p_{t|0}$ is defined only on $\mathcal{M}$, we evaluate $\Delta^{\mathcal{M}} p_{t|0}$ by choosing any smooth extension $\bar{p}_{t|0}$ to a neighborhood of $\mathcal{M}$ in $\tilde{\mathcal{M}}$. Any such extension can be used since the projection matrix $P(x)$ projects vectors into the tangent bundle of $\mathcal{M}$.

## A.2. Minakshisundaram–Pleijel recursion formula

We follow the notation in Section 4.3. The heat kernel has an $n$-th order parametrix expansion (Rosenberg, 1997):

$$\begin{aligned} p_t(x|x_0) &= G_t^{(n)}(x, x_0) + O(t^n), \\ G_t^{(n)}(x, x_0) &:= G_t(x, x_0)\Big(u_0(x, x_0) + u_1(x, x_0)t + \cdots, u_{n-1}(x, x_0)t^{n-1}\Big). \end{aligned} \tag{35}$$

where $u_0(x, x_0) = D_{x_0}(x)^{-1/2}$ and $u_1, \cdots, u_{n-1}$ can be computed recursively from earlier ones:

$$u_i(x, x_0) = r^{-i} D^{-1/2}(x) \int_0^r D^{1/2}(z(s)) \Delta_x^{\mathcal{M}} u_{i-1}(x, z(s)) s^{i-1} ds \tag{36}$$

where $r = d(x, x_0)$ and $z : [0, r] \to \mathcal{M}$ is the unit-speed geodesic from $x_0$ to $x$.

## A.3. Worked example of Minakshisundaram–Pleijel recursion formula on $S^2$

Since the recursion formula Equation (36) has an integral, it cannot be analytically computed in general. Instead, if the dimension of $\mathcal{M}$ is sufficiently small or the heat kernel only depends on a small number of variables, we can Taylor-expand the terms $u_i$ and compute the recursion formula, using a Computer Algebra System (CAS), e.g., sympy (Meurer et al., 2017).

Take $x_0 = [0, 0, 1]^T$ and for $x \in S^2 \subseteq \mathbb{R}^3$, let $r = \arccos(x^T x_0)$ be the geodesic distance between $x_0$ and $x$. Due to the symmetry of $p_{t|0}$, the heat equation only depends on the distance $r$. For a function $f = f(r) : S^2 \to \mathbb{R}$ that only depends on $r$, the Laplace-Beltrami operator reduces to:

$$\Delta^{S^2} f = f'' + \left(\frac{1}{r} + \frac{D'}{D}\right) f' \tag{37}$$

where $D(r) = \frac{\sin r}{r}$ is the volume distortion. The recursion formula Equation (36) reduces to:

$$u_i = r^{-i} D(r)^{-1/2} \int_0^r D^{1/2}(s) \Delta u_{i-1}(s) s^{i-1} ds \tag{38}$$

where $u_0 = D(r)^{-1/2}$. In the sympy environment, we conduct series expansion before computing the integral in $s$. Below we list some examples of computed terms $u_i$ using sympy:

$$
\begin{aligned}
u_1 &= \frac{9689 r^8}{319334400} + \frac{17 r^6}{56700} + \frac{31 r^4}{10080} + \frac{r^2}{30} + \frac{1}{3} \\
u_2 &= -\frac{1433 r^8}{139708800} + \frac{3277 r^6}{19958400} + \frac{19 r^4}{15120} + \frac{r^2}{105} + \frac{1}{15} \\
u_3 &= -\frac{9619 r^8}{1026432000} - \frac{5713 r^6}{52390800} + \frac{71 r^4}{110880} + \frac{r^2}{315} + \frac{4}{315}
\end{aligned}
\tag{39}
$$

where the series are truncated at 10-th order. The computed parametrix approximation up to third order is used for the initial condition of heat kernel, for $\mathcal{M} = S^2$ and $SO(3)$.

### A.4. Heat equation on $SPD(n)$ in log-eigenvalue coordinates

Let $\mathrm{SPD}(n) = \{X \in \mathbb{R}^{n \times n} : X = X^\top, \ X \succ 0\}$ equipped with the affine-invariant metric

$$g_X(U, V) \ = \ \mathrm{tr}\big(X^{-1} U X^{-1} V\big), \qquad X \in \mathrm{SPD}(n), \ U, V \in T_X \mathrm{SPD}(n) \cong \mathrm{Sym}(n). \tag{40}$$

The congruence action of $GL(n)$, $X \mapsto A X A^\top$, is isometric under (40). In particular, the heat kernel started at $I$ is conjugation-invariant: $p_t(Q X Q^\top \mid I) = p_t(X \mid I)$ for all $Q \in O(n)$, hence it depends only on the eigenvalues of $X$.

**Log-eigenvalue coordinates.** Write the spectral decomposition

$$X \ = \ Q \, \mathrm{diag}(\lambda_1, \ldots, \lambda_n) \, Q^\top, \qquad Q \in O(n), \ \lambda_i > 0,$$

and define log-eigenvalues $r_i := \log \lambda_i$, so that

$$X \ = \ Q \, \mathrm{diag}(e^{r_1}, \ldots, e^{r_n}) \, Q^\top. \tag{41}$$

For a conjugation-invariant function $f(X) = \bar{f}(r)$ (symmetric in $r$), the Laplace–Beltrami operator reduces to its radial part.

**Radial volume density.** In the coordinates $(Q, r)$, the Riemannian volume element factorizes as

$$d\mu_{\mathrm{SPD}}(X) \ = \ C \, J(r) \, dr \, d\mu_{\mathrm{Haar}}(Q), \qquad J(r) \ = \ \prod_{1 \le i < j \le n} \sinh\left(\frac{r_i - r_j}{2}\right), \tag{42}$$

where $C$ is a constant. It is often convenient to write

$$J(r) \ = \ \left( \prod_{i<j} \frac{r_i - r_j}{2} \right) D(r), \qquad D(r) := \prod_{i<j} \frac{\sinh\left(\frac{r_i - r_j}{2}\right)}{\frac{r_i - r_j}{2}}. \tag{43}$$

**Radial Laplacian.** For $f(X) = \bar{f}(r)$, the radial Laplace–Beltrami operator is

$$(\Delta f)(X) \ = \ (\Delta_{\mathrm{rad}} \bar{f})(r) \ := \ \frac{1}{J(r)} \sum_{i=1}^n \partial_{r_i}\big( J(r) \, \partial_{r_i} \bar{f}(r) \big). \tag{44}$$

Expanding (44) yields

$$\Delta_{\mathrm{rad}} \bar{f}(r) \ = \ \sum_{i=1}^n \partial_{r_i}^2 \bar{f}(r) \ + \ \sum_{i=1}^n \big( \partial_{r_i} \log J(r) \big) \, \partial_{r_i} \bar{f}(r). \tag{45}$$

Using (42),

$$\log J(r) = \sum_{i<j} \log \sinh\Big(\frac{r_i - r_j}{2}\Big) \quad \Longrightarrow \quad \partial_{r_i} \log J(r) = \frac{1}{2} \sum_{j\neq i} \coth\Big(\frac{r_i - r_j}{2}\Big).$$

Plugging this into (45) gives the explicit radial operator

$$\Delta_{\mathrm{rad}} \bar{f}(r) = \sum_{i=1}^{n} \partial_{r_i}^2 \bar{f}(r) + \frac{1}{2} \sum_{i\neq j} \coth\Big(\frac{r_i - r_j}{2}\Big) \partial_{r_i} \bar{f}(r). \tag{46}$$

Thus, the heat equation $\partial_t p_t = \Delta p_t$ becomes

$$\partial_t p_t(r) = \sum_{i=1}^{n} \partial_{r_i}^2 p_t(r) + \frac{1}{2} \sum_{i<j} \coth\Big(\frac{r_i - r_j}{2}\Big) \big(\partial_{r_i} - \partial_{r_j}\big) p_t(r). \tag{47}$$

after grouping the symmetric terms.

## B. Experiment Details

This section describes the experimental settings.

**Architectures and optimization.** Unless noted otherwise, for PINN training we use the modified MLP of Wang et al. (2023) with manifold-dependent embeddings. We implement the PINN in JAX (Bradbury et al., 2018), which enables computing higher-order derivatives while staying within practical GPU memory limits. For the denoiser, we use the same architecture as in the baseline experiments. All models are trained with the Adam optimizer and a cosine learning-rate schedule. The initial learning rate is tuned over $\{3 \cdot 10^{-3}, 1 \cdot 10^{-3}, 3 \cdot 10^{-4}, 1 \cdot 10^{-4}\}$.

**Hardware and training speed.** All models are trained on a single NVIDIA GeForce RTX 3090 Ti GPU. All experiments finish within one day, except training the diffusion model on QM9, which takes about 3 days.

### B.1. Climate science datasets on $S^2$

For PINN training, we fix the initial point $x_0 = [0, 0, 1]^T \in S^2$ and learn the heat kernel conditioned on $x_0$. For an arbitrary $x_0 \in S^2$, the heat kernel $p_{t|0}(\cdot \mid x_0)$ is recovered via an isometry that maps $x_0 \mapsto [0, 0, 1]^T$. The PINN is a 4-layer modified MLP with 256 hidden units and sinusoidal embeddings. We train the PINN on $[t_0, t_{\max}] = [0.1, 5]$, imposing the initial condition at $t = t_0$. The denoiser is a 5-layer MLP with 512 hidden units.

### B.2. Synthetic data on $SO(3)$

We view $SO(3)$ as the quotient manifold $S^3/\{\pm I\}$, where $S^3 \subset \mathbb{R}^4$ is the unit 3-sphere. The PINN and denoiser architectures are identical to Section B.1, except for input/output dimensions.

### B.3. Traffic analysis on $SPD(10)$

Motivated by the symmetric and antisymmetric terms in Equation (27), we introduce two permutation-invariant embeddings of $r$:

$$S_k(r) = \Big(\sum_i |r_i|^k\Big)^{1/k}, \quad A_k(r) = \Big(\sum_{i<j} |r_i - r_j|^k\Big)^{1/k}, \tag{48}$$

where $k$ is sampled uniformly from $[1, 5]$. Since $p_{t|0}$ is symmetric in $r = (r_1, \cdots, r_n)$, these embeddings provide a permutation-invariant representation. We combine them with a sinusoidal embedding of $t$ and feed the resulting features into the modified MLP PINN. We train the PINN on $[t_0, t_{\max}] = [0.1, 1]$, since the hyperbolic geometry of $SPD(n)$ can cause diffusion across scales even for small $t$. For the denoiser, we use SPD-Net introduced in Li et al. (2024). This architecture mimics the shape of U-Net (Ronneberger et al., 2015) with operations on $SPD(n)$.

*Table 8.* Runtime of the learned PINN heat-kernel surrogate. We report the average wall-clock time for log-density evaluation, score evaluation, and one MCMC-based forward sampling step with batch size 128.

| $\mathcal{M}$ | $\log p$ eval. (ms) | $\nabla \log p$ eval. (ms) | MCMC sample (ms) |
|---|---|---|---|
| $S^2$ | 0.116 | 0.178 | 1.6 |
| $SO(3)$ | 0.118 | 0.177 | 6.4 |
| $SPD(10)$ | 0.206 | 0.302 | 217.7 |
| $\mathbb{R}^{k \times n}/S_n$ | 1.488 | 3.404 | – |

### B.4. Brain connectivity (EEG) analysis on $SPD(n)$

The PINN architecture is identical to Section B.3. For the denoiser, we use a 4-layer MLP with 256 hidden units that takes the upper-triangular entries (including the diagonal) of an SPD matrix as input and outputs an upper-triangular matrix $U$ (including the diagonal). We then map $U \mapsto U^T U$ to ensure positive-definiteness.

### B.5. Molecule generation on $\mathbb{R}^{k \times n}/S_n$

Since the heat kernel is permutation-symmetric, we use permutation symmetry as an inductive bias for the PINN. For the initial point $X_0 = [x_0^{(1)}, \cdots, x_0^{(n)}]^T$ and a perturbed configuration $X = [x^{(1)}, \cdots, x^{(n)}]^T$, the PINN input is the matrix of relative distances $R = (R_{ij})_{1 \le i, j \le n}$, with $R_{ij} = |x^{(i)} - x_0^{(j)}|$. To encode permutation symmetry, we use a Graph Neural Network (GNN) (Wu et al., 2020), treating diagonal entries as node features and off-diagonal entries as edge features. Node and edge features are updated by MLPs with two hidden layers and 64 hidden units, and edge features are aggregated by summation. We stack two such GNN layers as the PINN.

For the denoiser, following the baseline (Hoogeboom et al., 2022), we use the EGNN architecture (Satorras et al., 2021) with 9 layers and 256 hidden units.

### B.6. Runtime analysis

We report the runtime overhead of the learned PINN heat-kernel surrogate. All timings are measured on a single NVIDIA GeForce RTX 3090 Ti GPU using the same implementation as in the main experiments. Since the PINN is implemented in JAX, all per-call timings are measured after JIT compilation. We evaluate the average wall-clock time for computing the learned log heat kernel $\log p_{t|0}^{(\theta)}$, its conditional score $\nabla \log p_{t|0}^{(\theta)}$, and one MCMC-based forward sampling step with batch size 128.

The cost of evaluating the learned log heat kernel and its score is small for all considered manifolds. The main additional overhead comes from MCMC-based forward sampling, especially on $SPD(10)$, where each proposal involves matrix operations under the affine-invariant geometry. The permutation-quotiented point-cloud experiment does not use the same MCMC forward-sampling procedure, so we omit its MCMC timing.

## C. Statements and proofs

### C.1. Stability under a zero PINN residual, with an approximate initializer

**Theorem C.1** (Stability of an initialized heat-kernel approximation). *Let $(\mathcal{M}, g)$ be a connected, complete $d$-dimensional Riemannian manifold without boundary, and let $\Delta$ be the Laplace–Beltrami operator. Let $p(t, x, y)$ denote the (minimal) heat kernel (Grigor'yan, 2012), i.e. the unique smooth function on $(0, \infty) \times \mathcal{M} \times \mathcal{M}$ such that for every $y \in \mathcal{M}$*

$$(\partial_t - \Delta_x)p(t, x, y) = 0, \qquad \lim_{t \downarrow 0} p(t, \cdot, y) = \delta_y \quad \textit{(in distributions)}, \qquad p(t, x, y) > 0,$$

*and satisfying the Chapman–Kolmogorov property*

$$p(t + s, x, z) = \int_{\mathcal{M}} p(t, x, y)\, p(s, y, z)\, d\operatorname{vol}_g(y), \qquad t, s > 0.$$

*Fix $x_0 \in \mathcal{M}$ and abbreviate*

$$p_{t|0}(x) := p(t, x, x_0).$$

*Fix $0 < t_0 < t_{\max}$ and $r_{\max} > 0$, and define the space–time domain*

$$D := [t_0, t_{\max}] \times B(x_0, r_{\max}), \qquad B(x_0, r_{\max}) = \{x \in \mathcal{M} : d_{\mathcal{M}}(x_0, x) \leq r_{\max}\}.$$

*Let $q_{t_0} : \mathcal{M} \to (0, \infty)$ be measurable and assume the* global relative initialization error *is uniformly small:*

$$\sup_{y \in \mathcal{M}} \left| \frac{q_{t_0}(y)}{p_{t_0|0}(y)} - 1 \right| \leq \varepsilon, \qquad \text{for some } \varepsilon \in [0, 1). \tag{49}$$

*Define $\tilde{p}(t, x)$ for $t \geq t_0$ by the heat semigroup representation*

$$\tilde{p}(t, x) := \int_{\mathcal{M}} p(t - t_0, x, y)\, q_{t_0}(y)\, d\operatorname{vol}_g(y).$$

*(Under (49) the integral is finite, since $q_{t_0} \leq (1 + \varepsilon)p_{t_0|0}$ and $\int p(t - t_0, x, y)p_{t_0|0}(y)\, dy = p_{t|0}(x) < \infty$.) Then $\tilde{p}$ is a (mild) solution of $\partial_t \tilde{p} = \Delta \tilde{p}$ on $(t_0, \infty) \times \mathcal{M}$, and is smooth for every $t > t_0$.*

*Then the following hold (stated only on $D$, although (1) in fact holds for all $x \in \mathcal{M}$).*

**(1) Propagation of the relative error (on $D$).** *For all $(t, x) \in D$,*

$$\left| \frac{\tilde{p}(t, x)}{p_{t|0}(x)} - 1 \right| \leq \varepsilon, \quad \text{equivalently} \quad |\tilde{p}(t, x) - p_{t|0}(x)| \leq \varepsilon\, p_{t|0}(x), \tag{50}$$

*and in particular $(1 - \varepsilon)p_{t|0}(x) \leq \tilde{p}(t, x) \leq (1 + \varepsilon)p_{t|0}(x)$ on $D$.*

**(2) Log-density stability (on $D$).** *For all $(t, x) \in D$,*

$$\left| \log \tilde{p}(t, x) - \log p_{t|0}(x) \right| \leq \max\{-\log(1 - \varepsilon),\, \log(1 + \varepsilon)\} \leq \frac{\varepsilon}{1 - \varepsilon}. \tag{51}$$

**(3) Score stability away from the initialization time (on $D_\delta$).** *Fix any $\delta \in (0, t_{\max} - t_0]$ and define*

$$D_\delta := [t_0 + \delta, t_{\max}] \times B(x_0, r_{\max}).$$

*Define*

$$\bar{A}_\delta := \sup_{\substack{t \in [t_0 + \delta, t_{\max}] \\ x \in B(x_0, r_{\max})}} \frac{1}{p_{t|0}(x)} \int_{\mathcal{M}} \|\nabla_x p(t - t_0, x, y)\|\, p_{t_0|0}(y)\, d\operatorname{vol}_g(y), \tag{52}$$

$$B_\delta := \sup_{\substack{t \in [t_0 + \delta, t_{\max}] \\ x \in B(x_0, r_{\max})}} \|\nabla_x \log p_{t|0}(x)\|. \tag{53}$$

*Assume $\bar{A}_\delta < \infty$ (this is an integrability/regularity condition on the heat kernel gradients; it holds under standard heat kernel gradient bounds, e.g. Li–Yau-type estimates (Li & Yau, 1986), on many complete manifolds). Then for all $(t, x) \in D_\delta$,*

$$\|\nabla_x \log \tilde{p}(t, x) - \nabla_x \log p_{t|0}(x)\| \leq \frac{\varepsilon}{1 - \varepsilon} \left( \bar{A}_\delta + B_\delta \right). \tag{54}$$

*Proof.* We use only positivity and the Chapman–Kolmogorov property.

**Step 1: Semigroup/Chapman–Kolmogorov formulas.** By definition,

$$\tilde{p}(t, x) = \int_{\mathcal{M}} p(t - t_0, x, y)\, q_{t_0}(y)\, dy, \qquad t \geq t_0.$$

By Chapman–Kolmogorov,

$$p_{t|0}(x) = p(t, x, x_0) = \int_{\mathcal{M}} p(t - t_0, x, y)\, p(t_0, y, x_0)\, dy = \int_{\mathcal{M}} p(t - t_0, x, y)\, p_{t_0|0}(y)\, dy.$$

Hence for $t \geq t_0$,

$$\tilde{p}(t, x) - p_{t|0}(x) = \int_{\mathcal{M}} p(t - t_0, x, y) \left( q_{t_0}(y) - p_{t_0|0}(y) \right) dy. \tag{55}$$

**Step 2: Proof of (1) (on $D$).**  Assumption (49) is equivalent to

$$|q_{t_0}(y) - p_{t_0|0}(y)| \le \varepsilon\, p_{t_0|0}(y) \qquad \text{for all } y \in \mathcal{M}.$$

Using (55), positivity of $p$, and linearity, for any $t \in [t_0, t_{\max}]$ and any $x \in \mathcal{M}$ (in particular $x \in B(x_0, r_{\max})$),

$$\begin{aligned}
|\tilde{p}(t,x) - p_{t|0}(x)| &\le \int_{\mathcal{M}} p(t - t_0, x, y)\, |q_{t_0}(y) - p_{t_0|0}(y)|\, dy \\
&\le \varepsilon \int_{\mathcal{M}} p(t - t_0, x, y)\, p_{t_0|0}(y)\, dy \\
&= \varepsilon\, p_{t|0}(x).
\end{aligned}$$

This yields (50) on $D$, and the two-sided bound $(1-\varepsilon)p_{t|0} \le \tilde{p} \le (1+\varepsilon)p_{t|0}$ follows immediately.

**Step 3: Proof of (2).**  From (1), for all $(t,x) \in D$ we have

$$1 - \varepsilon \le \frac{\tilde{p}(t,x)}{p_{t|0}(x)} \le 1 + \varepsilon.$$

Taking logs gives

$$\log(1 - \varepsilon) \le \log \tilde{p}(t,x) - \log p_{t|0}(x) \le \log(1 + \varepsilon),$$

hence (51). The final bound $\max\{-\log(1-\varepsilon), \log(1+\varepsilon)\} \le \varepsilon/(1-\varepsilon)$ is a standard calculus inequality for $\varepsilon \in [0,1)$.

**Step 4: Proof of (3): gradient bound for $\tilde{p} - p_{t|0}$.**  Fix $\delta > 0$ and consider $(t,x) \in D_\delta$ so that $s := t - t_0 \in [\delta, t_{\max} - t_0]$. For $s \ge \delta > 0$ the kernel $p(s,x,y)$ is smooth in $(x,y)$, and (by the assumption $\bar{A}_\delta < \infty$) the function $y \mapsto \|\nabla_x p(s,x,y)\|\, p_{t_0|0}(y)$ is integrable uniformly over $x \in B(x_0, r_{\max})$ and $s \in [\delta, t_{\max} - t_0]$. Therefore we may differentiate (55) under the integral sign to obtain

$$\nabla_x\big(\tilde{p}(t,x) - p_{t|0}(x)\big) = \int_{\mathcal{M}} \nabla_x p(s,x,y)\, \big(q_{t_0}(y) - p_{t_0|0}(y)\big)\, dy.$$

Using (49) and the definition (52),

$$\begin{aligned}
\|\nabla_x(\tilde{p}(t,x) - p_{t|0}(x))\| &\le \int_{\mathcal{M}} \|\nabla_x p(s,x,y)\|\, |q_{t_0}(y) - p_{t_0|0}(y)|\, dy \\
&\le \varepsilon \int_{\mathcal{M}} \|\nabla_x p(s,x,y)\|\, p_{t_0|0}(y)\, dy \\
&\le \varepsilon\, \bar{A}_\delta\, p_{t|0}(x), \qquad (t,x) \in D_\delta.
\end{aligned}$$

**Step 5: Convert to a score bound.**  For $(t,x) \in D_\delta$,

$$\nabla_x \log \tilde{p} - \nabla_x \log p_{t|0} = \frac{\nabla_x(\tilde{p} - p_{t|0})}{\tilde{p}} + \nabla_x \log p_{t|0}\, \frac{p_{t|0} - \tilde{p}}{\tilde{p}}.$$

Taking norms and using (1), we have $\tilde{p} \ge (1-\varepsilon)p_{t|0}$ and $|p_{t|0} - \tilde{p}| \le \varepsilon p_{t|0}$ on $D_\delta$. Hence,

$$\left\| \frac{\nabla_x(\tilde{p} - p_{t|0})}{\tilde{p}} \right\| \le \frac{\varepsilon\, \bar{A}_\delta\, p_{t|0}}{(1-\varepsilon)p_{t|0}} = \frac{\varepsilon}{1-\varepsilon}\bar{A}_\delta,$$

and, using the definition (53),

$$\left\| \nabla_x \log p_{t|0}\, \frac{p_{t|0} - \tilde{p}}{\tilde{p}} \right\| \le B_\delta\, \frac{\varepsilon p_{t|0}}{(1-\varepsilon)p_{t|0}} = \frac{\varepsilon}{1-\varepsilon}B_\delta.$$

Combining yields (54).

**Step 6: Finiteness of $B_\delta$.** Since $(\mathcal{M}, g)$ is complete, Hopf–Rinow implies the closed ball $B(x_0, r_{\max})$ is compact. Because $p_{t|0}(x)$ is smooth and strictly positive for $t \geq t_0 + \delta > 0$, the map $(t, x) \mapsto \|\nabla_x \log p_{t|0}(x)\|$ is continuous on the compact set $D_\delta$, hence $B_\delta < \infty$. This completes the proof. $\qquad\square$

*Remark* C.2 (On the role of $\bar{A}_\delta$). In the compact case one may bound $\|\nabla_x p(s, x, y)\| \leq A_\delta\, p(s, x, y)$ with a finite uniform constant by taking a supremum over $y \in \mathcal{M}$. On a non-compact manifold this uniform ratio typically fails (already on $\mathbb{R}^d$), so we replace it by the averaged quantity $\bar{A}_\delta$ in (52). On many classes of complete manifolds (e.g. under Ricci lower bounds plus standard heat kernel Gaussian/gradient estimates), $\bar{A}_\delta$ is finite on $D_\delta$.

### C.2. Stability under a nonzero PINN residual

Theorem C.1 analyzes the idealized setting where the approximate initial condition is evolved exactly under the heat equation. In practice, however, the learned PINN is only an approximate solver: it is trained to match the initial/boundary condition and to make the PDE residual small, but the residual is not identically zero. We therefore give a complementary stability statement showing that a uniformly small residual for the log heat equation leads to a controlled log-density error.

**Theorem C.3** (Stability under a nonzero log-PDE residual). *Let $Q = [t_0, t_{\max}] \times \Omega$ be a compact space-time domain, where $\Omega \subset M$ is a spatial domain with smooth boundary. Let $\partial_p Q$ denote the parabolic boundary,*

$$\partial_p Q = \big(t_0 \times \Omega\big) \cup \big([t_0, t_{\max}] \times \partial\Omega\big). \tag{56}$$

*When $\Omega = M$ and $M$ is compact without boundary, the lateral boundary term is absent. Let $p = e^\phi$ be the exact heat kernel on $Q$, so that*

$$\partial_t p = \Delta_M p, \qquad \phi = \log p. \tag{57}$$

*Equivalently, $\phi$ satisfies*

$$\mathcal{R}[\phi] := \partial_t \phi - \Delta_M \phi - |\nabla\phi|_g^2 = 0. \tag{58}$$

*Let $\hat{\phi}$ be a smooth PINN approximation and define $\hat{p} = e^{\hat{\phi}}$. Suppose that, for some $\varepsilon_R, \varepsilon_B \geq 0$,*

$$|\mathcal{R}[\hat{\phi}](t, x)| \leq \varepsilon_R, \qquad (t, x) \in Q, \tag{59}$$

*and*

$$|\hat{\phi}(t, x) - \phi(t, x)| \leq \varepsilon_B, \qquad (t, x) \in \partial_p Q. \tag{60}$$

*Then, for every $(t, x) \in Q$,*

$$e^{-\varepsilon_B - \varepsilon_R(t-t_0)} p(t, x) \leq \hat{p}(t, x) \leq e^{\varepsilon_B + \varepsilon_R(t-t_0)} p(t, x). \tag{61}$$

*Equivalently,*

$$|\hat{\phi}(t, x) - \phi(t, x)| \leq \varepsilon_B + \varepsilon_R(t - t_0) \leq \varepsilon_B + \varepsilon_R(t_{\max} - t_0). \tag{62}$$

*Proof.* A direct calculation gives the key identity

$$(\partial_t - \Delta_M)\hat{p} = \hat{p}\big(\partial_t \hat{\phi} - \Delta_M \hat{\phi} - |\nabla\hat{\phi}|_g^2\big) = \hat{p}\,\mathcal{R}[\hat{\phi}]. \tag{63}$$

Hence the residual bound implies

$$-\varepsilon_R \hat{p} \leq (\partial_t - \Delta_M)\hat{p} \leq \varepsilon_R \hat{p}.$$

Define the exponentially rescaled functions

$$u^+(t, x) = e^{-\varepsilon_R(t-t_0)}\hat{p}(t, x), \qquad u^-(t, x) = e^{\varepsilon_R(t-t_0)}\hat{p}(t, x).$$

Then $u^+$ and $u^-$ are respectively a subsolution and a supersolution:

$$(\partial_t - \Delta_M)u^+ \leq 0, \qquad (\partial_t - \Delta_M)u^- \geq 0. \tag{64}$$

On $\partial_p Q$, the boundary assumption gives

$$e^{-\varepsilon_B} p \leq \hat{p} \leq e^{\varepsilon_B} p.$$

Since $p$ exactly solves the heat equation, the parabolic comparison principle applied to Equation (64) yields

$$u^+(t, x) \leq e^{\varepsilon_B} p(t, x), \qquad u^-(t, x) \geq e^{-\varepsilon_B} p(t, x)$$

throughout $Q$. Rearranging gives the density bound

$$e^{-\varepsilon_B - \varepsilon_R(t-t_0)} p(t,x) \leq \hat{p}(t,x) \leq e^{\varepsilon_B + \varepsilon_R(t-t_0)} p(t,x). \tag{65}$$

Taking logarithms gives

$$|\hat{\phi}(t,x) - \phi(t,x)| \leq \varepsilon_B + \varepsilon_R(t-t_0),$$

which proves the claimed log-density bound. $\qquad\square$

