# OpenReview forum: "Riemannian Diffusion Models on General Manifolds via Physics-Informed Neural Networks"
_ICML.cc/2026/Conference — ICML 2026 regular_

### Official Review · Reviewer_X4fL · 2026-03-09

**Soundness:** 2
**Presentation:** 3
**Significance:** 3
**Originality:** 3
**Overall Recommendation:** 4
**Confidence:** 4

**Summary:**

This paper proposes a general framework for drift-free Riemannian diffusion models when the manifold heat kernel is not available in closed form. The key idea is to approximate the manifold heat kernel by solving the heat equation using a physics-informed neural network (PINN), initialized via a short-time asymptotic expansion. The learned surrogate is then used to perform forward noising and to compute scores for denoising score matching. The framework is applied to several manifolds including $S^2$, $SO(3)$, $SPD(n)$, and permutation-quotiented point clouds for molecular generation. Experiments evaluate the method on density estimation, conditional generation on SPD matrices, EEG connectivity generation, and QM9 molecule generation.

**Compliance With Llm Reviewing Policy:**

Affirmed.

**Final Justification:**

Please see my "Rebuttal Acknowledgement"

**Key Questions For Authors:**

- Theorem 4.1 analyzes stability of the heat equation with approximate initialization, but not the learned PINN approximation. Can the authors provide theoretical or empirical analysis relating PDE residual or network approximation error to the resulting score error?
- The molecule example models point clouds via $\mathbb{R}^{k \times n}/S_n$. Since the permutation action is not free at collisions, the quotient is not globally a smooth manifold. How is this handled in the proposed framework?
- Section 5.3 derives the heat kernel starting at the identity. How is $p_t(X \mid X_0)$ computed in practice for arbitrary $X_0$?
- Could the authors report direct approximation errors of the learned heat kernel (e.g., on $S^2$ or $SO(3)$) to better isolate the quality of the surrogate?

**Limitations:**

yes

**Strengths And Weaknesses:**

Strengths:
- The paper addresses an important limitation of current Riemannian diffusion models: reliance on tractable heat kernels, which restricts applicability to a small class of manifolds.
- The proposed framework is conceptually simple and general: approximate the heat kernel via a PDE solver (PINN) and use it within score-based generative modeling.
- The empirical evaluation spans several different manifolds and application domains, demonstrating the flexibility of the approach.
- The theoretical result (Theorem 4.1 / Appendix C) provides a stability guarantee for heat flow under approximate initialization, which justifies the use of short-time kernel approximations.

Weaknesses:
- The theoretical guarantee only applies to the *exact* heat equation with approximate initialization, not to the learned PINN approximation used in practice. The paper does not analyze the effect of PDE residual or neural approximation error on the resulting score estimates.
- The treatment of quotient manifolds is somewhat imprecise. For example, $\mathbb{R}^{k \times n}/S_n$ is not globally a smooth manifold because the permutation action is not free when points collide; the paper’s formulation implicitly ignores these singular configurations.
- In the $SPD(n)$ section, the derivation focuses on the heat kernel starting at the identity and radial functions in eigenvalue coordinates, but the generative model requires conditional kernels for arbitrary $X_0$. The reduction used in practice is not clearly explained.
- The empirical section demonstrates downstream task performance but does not directly evaluate the quality of the learned heat-kernel surrogate (e.g., approximation error relative to known kernels on $S^2$ or $SO(3)$).

---

> ### Author Rebuttal · Authors · 2026-03-31
>
> We thank the reviewer for their careful assessment and for recognizing the motivation, theoretical grounding, and applicability of the paper. We also appreciate their concerns regarding the practical PINN approximation, the quotient-manifold formulation, the $\mathrm{SPD}(n)$ implementation, and direct validation of the heat-kernel surrogate.
>
> ## **PINN error analysis (theoretical).**
>
> We agree that Theorem 4.1 is stated in an idealized zero-residual setting: it analyzes stability when the approximate initializer is evolved under the exact heat equation, whereas in the actual method we train a PINN for the log heat kernel by minimizing both the initializer loss and the PDE residual. However, this assumption is not essential. The theorem can be relaxed to the case where the PINN has a uniformly small residual, as follows. The proof uses positivity of the approximate solution and a time-integrated form of the heat equation.
>
> **Theorem (informal).** Let $\phi = \log p$, so that $\phi$ satisfies
> $R[\phi] := \partial_t \phi - \|\nabla \phi\|^2 - \Delta \phi = 0$.
> Suppose that, on the space-time domain $Q := [t_0,t_{\max}] \times B$,
> $|R[\tilde{\phi}]| \le \varepsilon$
> for some $\varepsilon > 0$, and that the boundary condition is also accurate up to $\varepsilon$, i.e.
> $|\tilde{\phi} - \phi| \le \varepsilon \qquad \text{on } \partial_p Q$.
> Then the approximation error remains controlled over the domain $Q$ as
> $|\tilde{\phi}(t,x) - \phi(t,x)| \le \varepsilon + \varepsilon (t-t_0) \le \varepsilon\bigl(1+t_{\max}-t_0\bigr)$.
>
> **Proof sketch.** Write $\tilde{p} = e^{\tilde{\phi}}$. A direct calculation gives
> $(\partial_t - \Delta)\tilde{p} = \tilde{p}\, R[\tilde{\phi}],$
> and therefore
> $-\varepsilon \tilde{p} \le (\partial_t - \Delta)\tilde{p} \le \varepsilon \tilde{p}.$
> Now define
> $u_+(t,x) := e^{-\varepsilon (t-t_0)} \tilde{p}(t,x), u_-(t,x) := e^{\varepsilon (t-t_0)} \tilde{p}(t,x)$,
> then we get
> $(\partial_t - \Delta)u_+ \le 0, (\partial_t - \Delta)u_- \ge 0$.
>
> Since $u_{\pm}$ satisfy the approximate boundary condition $u_+ \le e^{\varepsilon} p$ and $u_- \ge e^{-\varepsilon} p$, integrating the above inequalities over time gives
> $u_+(t,x) \le e^{\varepsilon} p(t,x),u_-(t,x) \ge e^{-\varepsilon} p(t,x)$ on $Q$.
>
> Rearranging this gives $e^{-\varepsilon(1+t-t_0)}\, p(t,x) \le \tilde{p}(t,x) \le e^{\varepsilon(1+t-t_0)}\, p(t,x)$, and hence the theorem holds. $\square$
>
> ## **Approximation quality of PINN.**
>
> See “Approximation quality of PINN” in our response to Reviewer eB7n.
>
> ## **Runtime analysis.**
>
> See “Runtime analysis” in our response to Reviewer znne.
>
> ## **Answer to key questions.**
>
> - **Singularity in $\mathbb{R}^{N \times k}/S_N$.**
>
> We thank the reviewer for pointing this out. Strictly speaking, $\mathbb{R}^{N \times k}/S_N$ is not globally a smooth manifold because the permutation action is not free when two or more points collide. A precise smooth-manifold formulation should therefore be stated on the open dense regular set $\mathcal{R} = \\{ X \in \mathbb{R}^{N \times k} \mid x^{(i)} \neq x^{(j)} \text{ for all } i \neq j \\}$ and its quotient $\mathcal{R}/S_N$. The collision configurations form a lower-dimensional singular set of measure zero, so they do not affect the generic case and are not encountered in our data. In practice, our method operates on permutation-invariant quantities and is therefore well defined on the regular part; near a singular configuration, the numerical output may be viewed as following one of the neighboring smooth branches. We will revise the paper to make this distinction explicit.
>
> - **Computational details in SPD(n).**
>
> We thank the reviewer for pointing out that this part was not sufficiently explained in the paper. For any $X_0 \in SPD(n)$, there is an isometry of $\mathrm{SPD}(n)$ that maps $X_0$ to $I$, given by $Y \mapsto X_0^{-1/2}YX_0^{-1/2}$. Using this isometry, we map the pair $(X, X_0)$ to $(\hat{X}, I)$, eigen-decompose the matrix $\hat{X}$, and compute $\log p(\hat{X}, I) = \log p(X, X_0)$. This whole process is differentiable and hence $\nabla \log p$ can also be computed. Note that the score function only has the ‘radial’ (or eigenvalue-part) direction, hence computing $\log p$ using eigenvalues is legitimate.

---

> > ### Author Rebuttal · Reviewer_X4fL · 2026-04-04
> >
> > The authors have addressed several key concerns, particularly clarifying the treatment of quotient manifolds and the implementation on SPD(n), and provided a reasonable extension of the theoretical analysis to account for PINN residuals. While the connection between PDE approximation error and score estimation could still be made more explicit, I believe the paper is now technically sound.

---

> > > ### Author Response · Authors · 2026-04-08
> > >
> > > We thank the reviewer for the thoughtful follow-up and for recognizing that the main concerns have been addressed. We would also like to note that the PDE approximation error is empirically observed to be small in the low-noise regime, which is the regime most relevant for accurate score estimation and denoising, suggesting that its practical effect is limited. A more explicit theoretical characterization of how PINN approximation error propagates to score error and sampling quality remains an important direction for future work.
> > >
> > > In light of the reviewer’s updated assessment that the concerns are fully resolved and that the paper is now technically sound, we would be grateful if the reviewer would consider revisiting the overall recommendation.

---

### Official Review · Reviewer_znne · 2026-03-11

**Soundness:** 3
**Presentation:** 3
**Significance:** 3
**Originality:** 3
**Overall Recommendation:** 4
**Confidence:** 3

**Summary:**

The paper presents a new Riemannian diffusion model. The main idea is to support manifolds where the heat kernel is not known a priori. The approach approximates the heat kernel by directly solving the heat equation using a physics-informed neural network (PINN). The method is theoretically grounded, and experiments on several manifolds are presented.

**Compliance With Llm Reviewing Policy:**

Affirmed.

**Final Justification:**

The rebuttal has addressed most of my concerns. While the paper has some weaknesses, I believe its strengths outweigh them.

**Key Questions For Authors:**

1. Can the authors provide evidence for the quality of the approximation of the heat kernel?

2. What is the overall run time and how it compares with the baselines?

**Limitations:**

The limitations of the method are well addressed in the Conclusions section.

**Strengths And Weaknesses:**

Strengths:

1. The topic is important and timely.

2. The paper addresses an important gap in existing methods: how to deal with manifolds without a closed-form expression of the heat kernel, significantly broadening the scope of Riemannian diffusion models.

3. The presented method is theoretically grounded, and the rationale and flow are clearly presented.

4. The experiments cover several applications and nicely demonstrate the benefits of the proposed method.

Weaknesses:

1. The computational complexity is higher than that of the alternatives, and a comprehensive comparison to baselines, for example to Riemannian flow matching, in terms of overall runtime could be added.

2. The accuracy of the heat kernel estimation by the PINN, especially in the higher-dimensional cases, is not directly demonstrated, but only indirectly through the performance on a downstream task.

3. Using a global coordinate system could be limiting, and the sensitivity to the choice of the global coordinate system could be discussed.

4. The dependency on the choice of embedding in the higher-dimensional cases could be addressed.

Other specific comments:

1. At the beginning of the Background section, “Definitions and Notations,” the tangent space
T_xM is not explicitly defined.

2. While the description of the rationale and flow in Section 4 is clear, the implementation details are not, and could be described in more detail.

3. Regarding the structure: as is, Section 5 is a bit unusual. Consider merging it with Section 6.

4. In the experimental study, including comparisons to more than one baseline would enhance the results.

---

> ### Author Rebuttal · Authors · 2026-03-31
>
> We thank the reviewer for their careful assessment and for recognizing the motivation, theoretical grounding, and experimental breadth of the paper. We also appreciate their main concerns regarding runtime, direct validation of the heat-kernel approximation, and sensitivity to coordinate and embedding choices.
>
> ## **Approximation quality of PINN.**
>
> See “Approximation quality of PINN” in our response to Reviewer eB7n.
>
> ## **Runtime analysis.**
>
> The PINNs are parameterized by shallow MLPs with manifold-specific embeddings. Our implementation is written in JAX, so after JIT compilation, forward sampling via MCMC and the computation of $\nabla \log p$ introduce only modest overhead during training. Here, we report the wall-clock time for a single forward sampling step and for computing $\nabla\log p$ with batch size 128.
>
> | Manifold| $\log p$ (ms) | $ \nabla \log p$ (ms) | MCMC sample (ms) |
> |---------------------|-----------:|--------------:|-----------------:|
> |$S^2$| 0.116| 0.178| 1.6|
> | $SO(3)$| 0.118| 0.177| 6.4|
> | $\mathrm{SPD}(10)$| 0.206| 0.302| 217.7|
> | $\mathbb{R}^{N \times k}/S_N$ | 1.488| 3.404| —|
>
> We further report the total wall-clock training time for the four experiments below. In most cases, our method has a training cost comparable to the original implementation. For the $\mathrm{SPD}$ (taxi) experiment, our method is in fact faster due to our JAX reimplementation of the architecture.
>
> | Method | $S^2$ (Climate) | $SO(3)$ | $\mathrm{SPD}$ (Taxi) | QM9 |
> |---|---:|---:|---:|---:|
> | Original | 1.1 hours | 0.56 hours | 21 hours | 3.0 days |
> | Ours | 1.8 hours | 0.75 hours | 17 hours | 3.1 days |
>
>
> ## **Answer to key questions.**
>
> - **Usage of a global coordinate system and dependency on the choice of embedding.**
>
> Although we present our method as a general framework for Riemannian score-based diffusion, we agree that some practical choices can affect performance. In particular, the use of a global coordinate system may introduce sensitivity to the chosen parametrization, and in higher-dimensional settings the ambient embedding can also matter. This is especially relevant for $SPD(n)$, whose dimension scales as $\mathcal{O}(n^2)$; for $n=13,15$, training a PINN on such domains typically requires additional inductive bias. We view this as a practical trade-off for enabling exact score-based diffusion on complex, high-dimensional manifolds, and a more systematic study of coordinate- and embedding-robust designs is an important direction for future work.
>
> - **Details and presentation.**
>
> We agree that the presentation can be strengthened, especially with respect to the implementation details of PINN training and diffusion model training. We will provide a more detailed description in the revised manuscript.

---

> > ### Author Rebuttal · Reviewer_znne · 2026-04-03
> >
> > I would like to thank the authors for their responses. They have addressed my main concerns, and I will maintain my positive score.

---

### Official Review · Reviewer_eB7n · 2026-03-11

**Soundness:** 3
**Presentation:** 3
**Significance:** 3
**Originality:** 2
**Overall Recommendation:** 3
**Confidence:** 3

**Summary:**

This paper studies Riemannian diffusion models for manifold-supported data. A key challenge in this setting is that training typically requires access to the manifold heat kernel and related quantities such as its gradients, which are generally unavailable in closed form beyond a few highly symmetric manifolds. To address this issue, the authors propose to approximate the corresponding heat equation using a physics-informed neural network (PINN). Concretely, given an explicit specification of the manifold, the authors first choose an appropriate global coordinate representation and derive the corresponding Fokker–Planck / heat equation. They then use a short-time asymptotic approximation to replace the Dirac initial condition, and train a PINN to approximate the log heat kernel. The resulting surrogate can be used both for forward noising and for conditional score evaluation in denoising score matching. To demonstrate the generality of the approach, the authors evaluate it on several different manifolds, including S^2,SO(3) and SPD(n).

**Compliance With Llm Reviewing Policy:**

Affirmed.

**Key Questions For Authors:**

1. The core contribution of the paper is the surrogate approximation of the manifold heat kernel. Could the authors provide more direct validation of the quality of this approximation, for example by reporting heat-kernel error, score error, or consistency analysis in settings where reference solutions or stronger approximations are available?

2. The paper mentions that the overall pipeline is computationally heavy and more complex than alternative approaches such as Riemannian flow matching. Could the authors provide clearer comparisons in terms of wall-clock time, memory usage, or training cost, and further explain under what kinds of application scenarios or data geometries the proposed method is more worthwhile to adopt compared with these alternatives?

3. In the EEG experiments on SPD(n)SPD(n)SPD(n), the proposed method is outperformed by DiffeoCFM, which weakens the persuasiveness of the method to some extent. Could the authors further analyze the reason for this, for example whether it is affected by data quality, or whether it reflects an inherent limitation of the heat-kernel-based approach itself?

4. In the quotient-manifold point-cloud experiments, the proposed method is intended to bypass the computational difficulty caused by permutation symmetry and to highlight the generality of the approach. Could the authors discuss more concretely the scalability of this surrogate method as the number of points increases, and at what scale it may become impractical?

**Limitations:**

yes

**Strengths And Weaknesses:**

The paper focuses on a real and important bottleneck in Riemannian diffusion modeling, namely how to conduct training and sampling on general manifolds where the heat kernel is unavailable in closed form. The main contribution of this paper is to provide an approach that does not rely on manifold-specific analytic formulas or spectral structures, but instead transforms heat-kernel approximation into a learnable PDE-solving problem. From the perspective of method structure, the paper presents a relatively systematic pipeline: choosing a coordinate representation, deriving the PDE, constructing a short-time initialization, training a PINN surrogate, and plugging it back into the diffusion training / sampling pipeline. This makes the overall work logically coherent and structurally complete, and gives it a certain degree of generality for extension to new manifolds.

The experimental coverage of the paper is also relatively broad. It does not stop at a single toy manifold, but instead involves S^2,SO(3) and SPD(n), and quotient-manifold point clouds at the same time. This supports the authors’ claim that the method serves as a general backend for manifold diffusion, rather than a one-off construction for a specific geometry. The experimental results are also relatively balanced overall: it outperforms RSGM on the S^2,SO(3) tasks, outperforms SPD-DDPM on the SPD(10) task, and is competitive but not the best on the EEG SPD task. This way of presenting the results in fact strengthens the credibility of the paper.

From the perspective of soundness, beyond engineering-level approximation techniques, the paper explicitly and carefully discusses the issue that the Dirac initial condition cannot be handled directly in numerical computation, and constructs the initial condition through parametrix / short-time asymptotic approximation, together with a stability argument showing that solving the heat-kernel equation from an approximate initial condition can still maintain controllable error at later times. The method therefore has relatively complete theoretical justification.

From the perspective of significance, the problem studied by the paper is itself important, and the method is in principle broadly applicable; however, the practical barrier to adopting this method may be relatively high. This is because the entire pipeline is computationally heavy, involving PINN training, higher-order automatic differentiation, and MCMC-based forward sampling. The paper itself also acknowledges that, compared with some alternative routes (such as Riemannian flow matching), the proposed method is more complex and computationally more expensive. Therefore, although I recognize its problem setting and technical completeness, the paper has not yet sufficiently explained under which regimes this method is more worthwhile to adopt in practice compared with these simpler alternative methods.

In terms of presentation, the overall writing quality is good and the structure is relatively clear, but some parts, especially the derivation of the manifold PDE and the instantiations on different manifolds, are still rather dense, which raises the reading barrier for readers who are not familiar with this area.

Overall, I believe this paper is technically solid, with a clear motivation, a complete methodological chain, and good cross-manifold applicability. At the same time, however, the methodological originality is relatively limited, and the current experiments could still be further strengthened with more direct validation of the core heat-kernel surrogate. In addition, the paper should explain more clearly the actual advantages and costs of the method compared with simpler alternatives.

---

> ### Author Rebuttal · Authors · 2026-03-31
>
> We thank the reviewer for their careful assessment and for recognizing the motivation, technical completeness, and broad applicability of the paper. We also appreciate their main concerns regarding direct validation of the heat-kernel surrogate and the practical cost of the method relative to simpler alternatives.
>
> ## **Approximation quality of PINN.**
>
> Thanks for raising this important issue. For the two manifolds $S^2$ and $\mathrm{SO}(3)$, for which the heat kernel is available via the method of Lou et al. (2023), we report the approximation errors of $\log p$ and $\nabla \log p$. Also, for all four manifolds, we report the boundary error and the PDE
>  residual error at various values of $t$. The PINN residual error is normalized by the largest term appearing in the equation, e.g., for a PDE of the form $A = B + C$, we report $\frac{|A-B-C|}{\max\\{|A|, |B|, |C|\\}}$. The boundary error is likewise normalized by the corresponding ground-truth value. As the tables below show, the resulting PINNs consistently achieve small normalized boundary and residual errors, suggesting that they satisfy the prescribed boundary condition and governing PDE to a high degree of accuracy. Moreover, on $S^2$ and $SO(3)$, where direct comparison is possible, the approximation errors of $\log⁡ p$ and $\nabla \log p$ are also small, providing empirical evidence that the learned PINNs accurately approximate the relevant heat kernel quantities. Note that the relative error of $\nabla \log p$ can become large when $t$ is large since the heat kernel converges to the uniform distribution as $t$ grows, hence $\lim_{t\rightarrow \infty} \nabla \log p = 0$.  We will add these empirical results to the appendix in the revision.
>
> - **Comparison with exact heat kernels.**
>
> ### $S^2$
>
> | $t$ | $\lvert \text{diff}(\log p) \rvert$ | $\frac{\lvert \text{diff}(\log p) \rvert}{\lvert \log p \rvert}$ | $\lvert \text{diff}(\nabla \log p) \rvert$ | $\frac{\lvert \text{diff}(\nabla \log p) \rvert}{\lvert \nabla \log p \rvert}$ |
> |---:|---:|---:|---:|---:|
> | 0.30 | 0.0054 | 0.0009 | 0.0280 | 0.0248 |
> | 0.50 | 0.0080 | 0.0032 | 0.0183 | 0.1844 |
> | 1.00 | 0.0016 | 0.0007 | 0.0027 | 0.0896 |
> | 2.00 | 0.0016 | 0.0006 | 0.0006 | 0.1234 |
> | 4.00 | 0.0016 | 0.0006 | 0.0002 | 2.2984 |
>
> ### $\mathrm{SO}(3)$
>
> | $t$ | $\lvert \text{diff}(\log p) \rvert$ | $\frac{\lvert \text{diff}(\log p) \rvert}{\lvert \log p \rvert}$ | $\lvert \text{diff}(\nabla \log p) \rvert$ | $\frac{\lvert \text{diff}(\nabla \log p) \rvert}{\lvert \nabla \log p \rvert}$ |
> |---:|---:|---:|---:|---:|
> | 0.30 | 0.0006 | 0.0002 | 0.7668 | 0.0039 |
> | 0.50 | 0.0055 | 0.0013 | 0.1706 | 0.0028 |
> | 1.00 | 0.0029 | 0.0007 | 0.0039 | 0.0017 |
> | 2.00 | 0.0058 | 0.0014 | 0.0000 | 0.3669 |
> | 4.00 | 0.0232 | 0.0054 | 0.0123 | 8.8089 |
>
> - **PINN residuals.**
>
> ### $S^2$
>
> | boundary $(t=0.3)$ | residual $(t=0.3)$ | residual $(t=0.5)$ | residual $(t=1)$ | residual $(t=2)$ | residual $(t=4)$ |
> |---|---:|---:|---:|---:|---:|
> | $1.39\times 10^{-4}$ | $3.73\times 10^{-3}$ | $2.85\times 10^{-2}$ | $1.85\times 10^{-2}$ | $1.80\times 10^{-2}$ | $1.23\times 10^{-2}$ |
>
> ### $\mathrm{SO}(3)$
>
> | boundary $(t=0.3)$ | residual $(t=0.3)$ | residual $(t=0.5)$ | residual $(t=1)$ | residual $(t=2)$ | residual $(t=4)$ |
> |---|---:|---:|---:|---:|---:|
> | $1.99\times 10^{-4}$ | $2.78\times 10^{-2}$ | $7.70\times 10^{-1}$ | $8.19\times 10^{-1}$ | $4.26\times 10^{-2}$ | $1.24\times 10^{-1}$ |
>
> ### $\mathrm{SPD}(10)$
>
> | boundary $(t=0.1)$ | residual $(t=0.1)$ | residual $(t=0.3)$ | residual $(t=0.5)$ | residual $(t=1)$ |
> |---|---:|---:|---:|---:|
> | $1.49\times 10^{-2}$ | $4.76\times 10^{-3}$ | $3.50\times 10^{-3}$ | $3.40\times 10^{-3}$ | $6.04\times 10^{-3}$ |
>
> ### $\mathbb{R}^{N\times k}/S_N$
>
> | boundary $(t=0.01)$ | residual $(t=0.01)$ | residual $(t=0.03)$ | residual $(t=0.1)$ | residual $(t=0.2)$ | residual $(t=0.4)$ |
> |---|---:|---:|---:|---:|---:|
> | $2.06\times 10^{-3}$ | $6.61\times 10^{-3}$ | $1.39\times 10^{-1}$ | $1.54\times 10^{-1}$ | $1.94\times 10^{-1}$ | $1.75\times 10^{-1}$ |
>
> ## **Runtime analysis.**
>
> See “Runtime analysis” in our response to Reviewer znne.
>
> ## **Significance of our work.**
>
> See “Significance of our work” in our response to Reviewer GPoG.
>
> ## **Comparison with DiffeoCFM (EEG experiments).**
>
> See “Comparison with DiffeoCFM (EEG experiments)” in our response to Reviewer GPoG.
>
> **References.**
>
> Lou et al. (2023). Scaling Riemannian diffusion models. NeurIPS.

---

### Official Review · Reviewer_GPoG · 2026-03-14

**Soundness:** 4
**Presentation:** 4
**Significance:** 3
**Originality:** 3
**Overall Recommendation:** 5
**Confidence:** 5

**Summary:**

This paper presents a diffusion model on Riemannian manifolds that uses a physics-informed neural network (PINN) to approximate the diffusion process (heat equation) on the manifold. Previously, diffusion models on Riemannian manifolds were restricted to approximations or "simple" manifolds (namely Riemannian symmetric spaces) where the heat kernel is more tractable. Experiments are conducted for problems on the sphere ($S^2$), 3D rotation group ($SO(3)$), symmetric positive-definite matrices ($SPD(n)$), and permutation-invariant Euclidean space ($\mathbb{R}^n/S_n$).

**Compliance With Llm Reviewing Policy:**

Affirmed.

**Final Justification:**

The proposed method develops diffusion models on a Riemannian manifold by using a PINN approximation to learn the heat kernel. After clarification by the authors in the rebuttal, the proposed model does have advantages and is applicable to a wider class of manifolds than was previously possible.

**Key Questions For Authors:**

* Is there a "killer application", i.e., a manifold that is not a symmetric space or that has not been possible to construct a diffusion model? This would strongly motivate the proposed work.
* Is it possible to compare to Lou et al.'s method at least for some cases, e.g., maybe $S^2$ and $SO(3)$? If so, how does the proposed approach compare?
* How much does performance degrade because of the PINN approximation (versus if we have analytic heat kernel available)? One way to possibly get at this would be to simply compare in Euclidean space versus a vanilla DDPM.

**Limitations:**

yes

**Strengths And Weaknesses:**

Strengths:
* The proposed method provides a way to train diffusion models on general Riemannian manifolds through a PINN approximation to the heat kernel. This should make diffusion models on non-standard manifolds possible.
* The paper is clearly written and the math derivations are clear and appear correct.
* The worked examples for the heat equation for $S^2$, $SO(3)$, and $SPD(n)$ in embedded Euclidean spaces that are used to train the PINN are nicely derived and explained.
* The experiments do show effectiveness and improved performance over the "original" RSGM.

Weaknesses:
* The main weakness of this paper is that the purported advantage of the proposed method, which is its applicability to a more general class of manifolds than previously possible, is only hypothetical. The experiments do not include any examples of manifolds that could not already be accomplished by existing approaches, in particular, the method of Lou et al., 2023, where analytic evaluation of the heat kernel is possible for symmetric spaces. For example, the sphere ($S^2$), rotation group ($SO(3)$), and symmetric positive-definite matrices ($SPD(n)$) are all symmetric spaces. The permutation-quotiented Euclidean space is perhaps the best example, although it is testing more of a combinatorial/topological problem - the curvature of that space is flat. Furthermore, it does not prove the point that the proposed method makes diffusion models on more general manifolds possible because there are existing equivariant models for this space (Le et al., 2024; Irwin et al., 2025), which presumably work very well (the present paper does not compare to them, as they more parameters and are more sophisticated for that particular problem.)
* Given that several "simple" symmetric spaces are used in the experiments, is it possible to compare with the method of Lou et al.? It seems that it would be, and it is curious that such a comparison is not included. Lou et al. does have code. (Although, I expect $SPD(n)$ would be more difficult to implement, as this is not an example in that paper, but at least $S^2$ should be doable.) Such a comparison would be important to test how much having the PINN approximation matters versus having more exact evaluation of the heat kernel.
* The proposed method does not perform as well as DiffeoCFM on the $SPD(n)$ manifold, and the authors conclude that the data-driven diffeomorphic mapping to Euclidean space may just work better than the affine-invariant Riemannian geometry. I think this is probably the wrong conclusion. DiffeoCFM is using a log-Euclidean metric on $SPD(n)$ (after mean centering), which is a valid Riemannian geometry that is isometric to Euclidean space. This means that the heat kernel is now just the standard one and a standard diffusion model should work. The proposed method has the disadvantage that the PINN heat kernel is approximating, and that fact may explain its performance.
* (Minor point / correction) The authors mistakenly claim the work of (Lou et al., 2023) is only applicable to quotients of compact Lie groups (paragraph below equation 11). However, that work applies to symmetric spaces, which are always quotients of Lie groups $G/K$, where $K$ is compact, but $G$ is not necessarily compact. In fact, the space of symmetric positive-definite matrices $SPD(n) = GL^+(n) / SO(n)$ used in this paper is a symmetric space, and $GL^+(n)$ is not compact.

---

> ### Author Rebuttal · Authors · 2026-03-31
>
> We thank the reviewer for their careful reading and for recognizing the clarity of the paper. We also appreciate their concern about whether our work demonstrates advantages beyond existing approaches.
>
> ## **Applicability of our method (compared to Lou et al., 2023).**
>
> **Lou et al. (2023) assume the manifold is compact. This assumption is essential.** In the first paragraph of section 2.2, they explicitly state:
> > To generalize diffusion models to d-dimensional Riemannian manifolds M, which we assume to be compact, connected, and isometrically embedded in Euclidean space, one adapts the existing machinery to the geometrically more complex space.
>
> The requirement of compactness is fundamental in the method of Lou et al. (2023). They use spectral decomposition for the heat kernel expansion, which is generally available only when the manifold $\mathcal{M}$ is compact. Moreover, they resort to the representation theory of Lie algebras. The key property here is the semisimplicity of the Lie algebra; the Lie algebra of a Lie group is semisimple if and only if the Lie group is either Euclidean or compact. Therefore, their method does not provide a way to compute or approximate the heat kernel on $\mathrm{SPD}(n)$.
>
> “Our method enables the construction of Riemannian diffusion models on $\mathrm{SPD}(n)$ and $\mathbb{R}^{k\times N} / S_N$.” The heat kernels on these manifolds cannot be computed or approximated using the method of Lou et al. (2023). $SPD(n)$ has dimension $\mathcal{O}(n^2)$, so $\mathrm{SPD}(n)$ with $n=13$ or $15$ is high-dimensional and geometrically complex. The same is true for the quotient space $\mathbb{R}^{k\times N} / S_{N}$. Our work provides a unified framework for building Riemannian diffusion models on such manifolds.
>
> ## **Significance of our work.**
>
> Score-based diffusion models are among the most successful and widely adopted frameworks in generative modeling. However, compared with their Euclidean counterparts, Riemannian score-based diffusion models remain much less explored, especially in real-world-scale settings. The absence of a closed-form heat kernel is the central bottleneck in applying them to general Riemannian manifolds — a limitation consistently noted in the literature (De Bortoli et al. (2022), Mangoubi et al. (2025), Li et al. (2024), Collas et al. (2025)). While Lou et al. (2023) made progress on heat kernel approximation, their approach relies on strong assumptions such as compactness and symmetry of the manifold. One notable successful example is the SE(3) diffusion model (Yim et al., 2023), which was feasible precisely because the heat kernel on SO(3) (known as $IGSO(3)$) happens to be available in closed form. The key point is that this is the exception, not the rule: for the vast majority of practically relevant manifolds, no such closed form exists, and existing methods either do not apply or degrade substantially. Our work directly addresses this gap by providing a general, principled approach to heat kernel approximation that does not require algebraic symmetry or compactness.
>
> ## **Approximation quality of PINN.**
>
> See “Approximation quality of PINN” in our response to Reviewer eB7n. It includes the explicit comparison with the method of Lou et al. (2023).
>
> ## **Comparison with DiffeoCFM (EEG experiments).**
>
> In the EEG experiments, we follow the experimental protocol of Collas et al. (2025), including their evaluation procedure. In particular, they report both a *classification metric* and a *quality metric*. To compute these metrics, the SPD matrices are first normalized using the Riemannian mean of the training data, and then **the upper-triangular entries are flattened and treated as vectors in Euclidean space $\mathbb{R}^{n(n+1)/2}$**. Both the classification and quality metrics are therefore computed in this Euclidean representation, rather than directly on the SPD manifold. As a result, these evaluation metrics are inherently Euclidean-oriented and may not fully reflect the geometric advantages of our method.
>
> ## **Answer to key questions.**
>
> - **Is there a "killer application"?**
>
> We believe the strongest current applications are $\mathrm{SPD}(n)$ data and permutation-invariant set-valued data in $\mathbb{R}^{k\times N}/S_N$, where intrinsic score-based diffusion is difficult precisely because the heat kernel is not available in a tractable form.
>
> **References.**
>
> Collas et al. (2025). Riemannian flow matching for brain connectivity matrices via pullback geometry. arXiv:2505.18193.\
> De Bortoli et al. (2022). Riemannian score-based generative modelling. NeurIPS.\
> Li et al. (2024). SPD-DDPM: Denoising diffusion probabilistic models in the symmetric positive definite space. AAAI.\
> Lou et al. (2023). Scaling Riemannian diffusion models. NeurIPS.\
> Mangoubi et al. (2025). Efficient diffusion models for symmetric manifolds. arXiv:2505.21640.\
> Yim et al. (2023). SE(3) diffusion model with application to protein backbone generation. arXiv:2302.02277.

---

> > ### Author Rebuttal · Reviewer_GPoG · 2026-04-07
> >
> > Thanks for the response, and thank you for clarifying for me that Lou et al. is only applicable to compact symmetric spaces (which I didn't realize). Given this, plus the evaluation of the PINN accuracy, I will raise my score.

---

### Decision · Program_Chairs · 2026-04-30

**Decision:**

Accept (regular)

**Comment:**

This paper has generally positive reviews. The key points raised are as follows.

Strengths:
- Important topic and area
- Compared to other papers in this literature, a wider class of examples is presented, including ones that are of relatively high dimension such as SPD(10).
- The mathematical aspects are clearly presented

Limitations:
- Examples considered are primarily analysis manifolds and already covered by existing approaches, no manifolds represented by meshes or other manifolds without a clean mathematical description are considered.
- Methodological originality is relatively limited, given both PINNs and Riemannian diffusion models are well-developed.

On balance of these examples, I think this work is over the bar, but not massively so, so I am recommending acceptance but would not mind if I was overruled. The main thing that would have convinced me to decisively recommend acceptance is if at least one manifold which does not come with a clean mathematical description was included. In addition, I would like to make the authors aware of additional works which allow one to compute the heat kernel on manifolds including some of the ones they are using: "Stationary Kernels and Gaussian Processes on Lie Groups and their Homogeneous Spaces I: the compact case", Azangulov et al., JMLR 2024 - which provides explicit representation-theoretic formulas for computing heat kernels on manifolds.